# Neural Posterior Estimation with Latent Basis Expansions

**Declan McNamara, Yicun Duan, Jeffrey Regier**
Department of Statistics
University of Michigan
Ann Arbor, MI 48109, USA
`{declan, pduan, regier}@umich.edu`

## Abstract

Neural posterior estimation (NPE) is a likelihood-free amortized variational inference method that approximates projections of the posterior distribution. To date, NPE variational families have been either simple and interpretable (such as the Gaussian family) or highly flexible but black-box and potentially difficult to optimize (such as normalizing flows). In this work, we parameterize variational families via basis expansions of the latent variables. The log density of our variational distribution is a linear combination of latent basis functions (LBFs), which may be fixed a priori or adapted to the problem class of interest. Our training and inference procedures are computationally efficient even for problems with high-dimensional latent spaces, provided only a low-dimensional projection of the posterior is of interest, owing to NPE's automatic marginalization capabilities. In numerous inference problems, the proposed variational family outperforms existing variational families used with NPE, including mixtures of Gaussians (mixture density networks) and normalizing flows, and also outperforms an existing basis expansion method for variational inference.

## 1 Introduction

Neural Posterior Estimation (NPE) is an increasingly popular approach to Bayesian inference (Papamakarios & Murray, 2016; Cranmer et al., 2020; Dax et al., 2021; Ward et al., 2022). In NPE, a neural network is trained exclusively on synthetic data—latent variables drawn from the prior paired with observations—to learn the inverse mapping from observations to latent variables. Once trained, this network can produce posterior approximations for real data in a single forward pass. In contrast to traditional (ELBO-based) variational inference, NPE does not require likelihood evaluations. Furthermore, when the generative model contains both parameters of interest and nuisance variables, NPE can automatically marginalize over the nuisance parameters during training: by simulating complete data and then discarding the nuisance variables to create training pairs, the method infers posterior projections for the parameters of interest (Ambrogioni et al., 2019).

Despite these advantages, NPE shares with traditional ELBO-based variational inference a fundamental trade-off between the flexibility of the variational family and the tractability of optimization. Simple variational families, such as Gaussians, enable stable optimization but often lack the expressiveness needed for complex posterior geometries. More flexible alternatives, such as Gaussian mixture models and normalizing flows, offer greater flexibility but can create difficult optimization landscapes with shallow local optima. Recent theoretical work (McNamara et al., 2024a) has established conditions for global convergence in NPE that hold for simple Gaussian variational families, but these results do not extend to the more expressive families commonly deployed in practice.

In this work, we propose a variational family specialized for NPE that leverages NPE's automatic marginalization and likelihood-free nature. Unlike standard variational inference, where nuisance variables must be modeled in the variational distribution, NPE applications typically require posteriors over just a few scientifically relevant parameters: high-dimensional posterior samples do not directly aid in interpretation but must instead be post-processed to estimate low-dimensional interpretable quantities. Because NPE does not require likelihood evaluations, this post-processing can often be

incorporated directly into the Bayesian model, resulting in a low-dimensional latent space of interest. In these low-dimensional settings, even numerical integration is computationally feasible, freeing us from the usual requirement of closed-form normalization.

We leverage this freedom by parameterizing the log density of variational distributions through latent basis expansions: a neural network processes observations to produce coefficients for linear combinations of basis functions over the latent space. This approach yields distributions in the exponential family—among the most flexible classes available (Pacchiardi & Dutta, 2022; Khemakhem et al., 2020; Sriperumbudur et al., 2017)—while maintaining favorable optimization properties. The resulting method, which we refer to as Latent Basis Function NPE (LBF-NPE), optimizes over the class of all exponential families of a fixed dimension $K$ by adaptively fitting the basis functions, denoted $s_\psi(z) \in \mathbb{R}^K$. Simultaneously, amortization is performed by fitting a separate network $f_\phi(x) \in \mathbb{R}^K$ that maps observations $x$ to coefficients of these basis functions (Section 3).

In Section 4, we introduce and analyze several variants of LBF-NPE. For a variant with fixed basis functions, such as B-splines or wavelets, optimization is convex despite the log-normalizer, providing stable training that has proven elusive for more complex variational families, and ensuring stable convergence to global optima under the conditions presented by McNamara et al. (2024a). Alternatively, both coefficients $f_\phi(x)$ and basis functions $s_\psi(z)$ can be fitted jointly through alternating optimization that exploits marginal convexity in each component. We employ stereographic projection reparameterization to address identifiability issues in this adaptive setting, constraining outputs to the unit hypersphere and stabilizing training.

We demonstrate superior performance of LBF-NPE across diverse inference problems, from synthetic benchmarks to real scientific applications (Section 6). LBF-NPE consistently converges to global optima on multimodal problems where mixture density networks converge to shallow local minima. LBF-NPE with just 20 basis functions achieves order-of-magnitude improvements in KL divergence over both MDNs and normalizing flows on complex 2D posteriors. The method successfully captures multimodal posteriors in astronomical object detection and substantially outperforms MDN baselines on cosmological redshift estimation using the LSST DESC DC2 survey dataset.

## 2 BACKGROUND

### 2.1 ELBO-BASED VARIATIONAL INFERENCE

In variational inference (VI), numerical optimization is used to select an approximation $q(z)$ of the posterior distribution of some model $p(z, x)$ on observables $x$ and latent variables $z$. The most common variational objective, the evidence lower bound (ELBO), targets minimization of the reverse KL divergence (Blei et al., 2017; Zhang et al., 2019; Kingma & Welling, 2019) by constructing the variational quantity

$$\text{ELBO}(\eta) := \mathbb{E}_{q(z;\eta)} \log \left( \frac{p(z, x)}{q(z; \eta)} \right) \leq \log p(x) \tag{1}$$

and performing maximization in $\eta$ for a fixed choice of $x$. Optimizing the ELBO remains a long-standing challenge in VI. Firstly, even simple variational families, such as the Gaussian, can exhibit problematic optimization landscapes without careful parameterization. Targeting the ELBO, even in the non-amortized setting, is generally a nonconvex problem (Liu et al., 2023; Domke, 2020; Domke et al., 2023). Secondly, and most significantly in our setting, ELBO-based variational inference does not provide a way to marginalize over nuisance latent variables $\xi$. For a model $p(z, \xi, x)$ on latent variables $\{z, \xi\}$ and observed variables $x$, inference on $z$ conditional on $x$ is a common target of interest. Yet ELBO-based methods typically cannot compute the quantity $p(z, x) = \int p(z, \xi, x)d\xi$ required to target the objective function (1), and instead typically require explicitly modeling the nuisance variables.

### 2.2 NEURAL POSTERIOR ESTIMATION

In amortized variational inference, the shared parameters of a deep neural network define variational approximations for arbitrary observations $x$ (Ganguly et al., 2024; McNamara et al., 2024a). Precisely, a variational approximation for latent variable $z$ is given by $q(z; \eta)$ with parameters $\eta$. Rather than fitting these parameters separately for each $x$, the amortized approach sets $\eta = f_\phi(x)$ for a neural

network $f_\phi$ with parameters $\phi$. This inference network, once fit, yields the variational posterior $q(z; f_\phi(x))$ for arbitrary $x$ by a single forward pass (Kingma & Welling, 2019; Ambrogioni et al., 2019). Neural posterior estimation (NPE) (Papamakarios & Murray, 2016) targets an expectation of the forward KL divergence for amortized VI:

$$\tilde{\mathcal{L}}_{\text{NPE}}(\phi) = \mathbb{E}_{p(x)}\text{KL}\left(p(z \mid x) \mid\mid q(z \mid x)\right). \tag{2}$$

Here, the integral over $p(x)$, the marginal of the model $p(z, \xi, x)$, indicates that the NPE objective averages over all possible draws from the generative process instead of averaging over observations $x$ from some finite training set. The objective is equivalent (up to a constant) to

$$\mathcal{L}_{\text{NPE}}(\phi) = -\mathbb{E}_{p(z,x)} \log q(z; f_\phi(x)), \tag{3}$$

where $q_\phi(z; \eta) \equiv q(z; f_\phi(x))$ for any $x$. Equation (3) admits unbiased estimation of its gradient with stochastic draws $(z, x) \sim p(z, \xi, x)$ from the joint model, readily obtained by ancestral sampling, even in the presence of nuisance latent variables $\xi$. For example, in a hierarchical model we can sample $(z, x) \sim p(z, x)$ by simulating an entire sequence $p(z)p(\xi_1 \mid z)p(\xi_2 \mid \xi_1) \cdots p(\xi_L \mid \xi_{L-1})p(x \mid \xi_L)$ and discarding variables that are not the target for inference. The expected forward KL objective has been independently derived and analyzed in several related works (Bornschein & Bengio, 2015; Ambrogioni et al., 2019).

## 3 LBF-NPE: BASIS EXPANSIONS FOR AMORTIZED LOG DENSITY ESTIMATION

We propose an amortized method to fit complex multimodal variational distributions, called Latent Basis Function NPE (LBF-NPE). For latent variables $z$ in latent space $I$, the method fits a basis function network $s_\psi : I \mapsto \mathbb{R}^K$, which evaluates $K$ basis functions $[s_\psi^{(1)}, \ldots, s_\psi^{(K)}]$ at any point $z$. The inference network $f_\phi : x \mapsto \eta \in \mathbb{R}^K$ maps observations to coefficients of the basis functions. The number of basis functions $K$ is fixed ahead of time, and larger $K$ may be used to increase expressivity (see Appendix E). The variational parameters to be fit are the neural network weights $\phi$ and $\psi$ for the networks $f_\phi$ and $s_\psi$. Below, we first construct an exponential family defined by the basis functions (Section 3.1) and then give the fitting routine for $f_\phi$ and $s_\psi$ (Section 3.2).

### 3.1 THE VARIATIONAL FAMILY

Fix $x \in \mathcal{X}$. Let $I \subseteq \mathbb{R}^d$ denote the latent space and let $\{s_\psi^{(i)}(z)\}_{i=1}^K$ be a collection of basis functions defined by parameters $\psi$. Selecting more basis functions (larger $K$) leads to a more expressive variational distribution, but also a higher-dimensional optimization problem. For a value $z \in I$, let $s_\psi(z) = [s_\psi^{(1)}, \ldots, s_\psi^{(K)}]^\top \in \mathbb{R}^K$. Our variational family is parameterized by coefficients $\eta \in \mathbb{R}^K$, and has the density function

$$q(z; \eta) \propto h(z) \exp\left(\eta^\top s_\psi(z)\right). \tag{4}$$

We aim to select $\eta$ and $\psi$ such that $q(z; \eta) \approx p(z \mid x)$. The log density is

$$\log q(z; \eta) = \log h(z) + \eta^\top s_\psi(z) - C, \tag{5}$$

where $C$ is the log of the normalizing constant. This variational family (Equation 4) is an exponential family (Wainwright & Jordan, 2008; Srivastava et al., 2014). The vector of basis functions $s_\psi(z)$ is the sufficient statistic, $\eta$ is the natural parameter vector, and $h(z)$ is any finite base measure on the latent space $I$ (i.e., $\int_I h(z)dz$ is finite).

We represent the $K$-dimensional quantity $s_\psi(z)$ as the output of a deep neural network with input $z$. Because the number and form of the basis functions $s_\psi(z)$ are arbitrary, the expressivity of this variational family is far greater than that of "classical" exponential families, such as the Gaussian family. As $K \to \infty$, the set of *all* exponential family distributions is arbitrarily rich: any distribution can be represented as an infinite-dimensional exponential family distribution (Khemakhem et al., 2020; Sriperumbudur et al., 2017).

### 3.2 THE AMORTIZED VARIATIONAL OBJECTIVE & GRADIENT ESTIMATOR

The formulation of Section 3.1 is *non-amortized*: it requires selecting a single $\eta$ for $q(z; \eta)$ to approximate the posterior for a single $x$. In NPE, we consider the amortized problem, where we define the posterior for an arbitrary $x$. We set $\eta = f_\phi(x)$, and thus require fitting two separate networks, $f_\phi$ and $s_\psi$. We fit the variational parameters, $\phi$ and $\psi$, by minimizing the NPE objective function, as given in general in Section 2.2. Our specific variational objective (up to constants) is thus

$$L_{\text{LBF-NPE}}(\phi, \psi) = -\mathbb{E}_{p(z,x)} \left( f_\phi(x)^\top s_\psi(z) - \log \left( \int \exp \left( f_\phi(x)^\top s_\psi(\tilde{z}) \right) h(\tilde{z}) \, d\tilde{z} \right) \right), \quad (6)$$

where the log-normalizer takes the form of the log of an integral with respect to the base measure $h$. The log-normalizer cannot be estimated without bias by Monte Carlo sampling due to the Jensen gap (Adil Khan et al., 2015). For training, we only require stochastic gradients, which can be computed using importance sampling. Focusing on the log-normalizer for now, we let $k_{\phi,\psi}(x, z) = f_\phi(x)^\top s_\psi(z)$. We suppress the dependence on $x$ for now, as it is fixed in the integral. Let $J(\phi, \psi) := \int \exp k_{\phi,\psi}(\tilde{z}) dh(\tilde{z})$. The gradient can be computed by estimating an expectation with respect to an exponentially tilted transformation of $h$. Let $\nabla_{\phi,\psi}$ denote the gradient with respect to either $\phi$ or $\psi$. Then, we have

$$\nabla_{\phi,\psi} \log J(\phi, \psi) = \frac{1}{J(\phi, \psi)} \cdot \nabla_{\phi,\psi} J(\phi, \psi) \tag{7}$$

$$= \frac{1}{J(\phi, \psi)} \cdot \int [\nabla_{\phi,\psi} k_{\phi,\psi}(\tilde{z})] \cdot \exp k_{\phi,\psi}(\tilde{z}) h(\tilde{z}) d\tilde{z} \tag{8}$$

$$= \int [\nabla_{\phi,\psi} k_{\phi,\psi}(\tilde{z})] \cdot \left( \frac{\exp k_{\phi,\psi}(\tilde{z}) h(\tilde{z})}{\int \exp k_{\phi,\psi}(z') h(z') dz'} \right) d\tilde{z} \tag{9}$$

$$=: \int [\nabla_{\phi,\psi} k_{\phi,\psi}(\tilde{z})] \cdot q_{\phi,\psi}(\tilde{z}) d\tilde{z} \tag{10}$$

where we recognize an expectation with respect to $q_{\phi,\psi}$, an exponentially tilted density that depends on the current values of $\phi, \psi$. This integral can be estimated by the use of self-normalized importance sampling (SNIS) with a proposal distribution $r(\tilde{z})$ (Owen, 2013):

$$\int [\nabla_{\phi,\psi} k_{\phi,\psi}(\tilde{z})] q_{\phi,\psi}(\tilde{z}) d\tilde{z} \approx \sum_{j=1}^{P} \frac{[\nabla_{\phi,\psi} k_{\phi,\psi}(\tilde{z}_j)] w(\tilde{z}_j)}{\sum_{k=1}^{P} w(\tilde{z}_k)},$$

where $w(z) = \frac{\exp(k_{\phi,\psi}(z)) h(z)}{r(z)}$ and $\tilde{z}_1, \ldots, \tilde{z}_P \overset{iid}{\sim} r$. The gradient estimator is thus biased for the true gradient, similar to other gradient estimators targeting the forward KL from the family of "wake-sleep" algorithms, but consistent as $P \to \infty$ (Le et al., 2019; Bornschein & Bengio, 2015; McNamara et al., 2024b).

Algorithm 1 details our gradient computation procedure.

## 4 VARIANTS & PROPERTIES OF LBF-NPE

We now motivate the construction of the LBF-NPE variational family by examining aspects of its optimization routine. As a result of parameterizing and targeting the log-density (cf. 5, 7), both our construction and optimization routine depend entirely on the inner product $f_\phi(x)^\top s_\psi(z)$. We elaborate on some key properties and variants of LBF-NPE that stem from this observation.

### 4.1 AFFINE GRADIENTS

In Section 3.2, we showed that the gradient of the objective for LBF-NPE depends only on the gradient of $k_{\psi,\phi}(x, z) = f_\phi(x)^\top s_\psi(z)$, via the relation

$$\nabla_{\phi,\psi} L_{\text{LBF-NPE}}(\phi, \psi) = -\mathbb{E}_{p(z,x)} \left[ \nabla_{\phi,\psi} k_{\phi,\psi}(x, z) - \mathbb{E}_{q_{\phi,\psi}(\tilde{z})} [\nabla_{\phi,\psi} k_{\phi,\psi}(x, \tilde{z})] \right] \tag{11}$$

---

**Algorithm 1:** Gradient Computation for LBF-NPE

---

**Inputs:** Sampling model $p(z, x)$; networks $f_\phi$ and $s_\psi$; proposal distribution $r(z)$.

Sample batch $\{(z_i, x_i)\}_{i=1}^B \overset{iid}{\sim} p(z, x)$

```
/* Gradient for log-normalizer */
```

Propose $\tilde{z}_1, \ldots, \tilde{z}_P \sim r$

Compute $k_j^i := k_{\phi,\psi}(x_i, \tilde{z}_j) = f_\phi(x_i)^\top s_\psi(\tilde{z}_j), \quad i \in [B], j \in [P]$

Compute unnormalized weights $w_j^i = \exp\left(k_j^i\right) \cdot h\left(\tilde{z}_j\right), \quad i \in [B], j \in [P]$

Compute $U_{\phi,\psi}^i = \sum_{j=1}^P \frac{w_j^i \nabla_{\phi,\psi} k_j^i}{\sum_{j'} w_{j'}^i}, \quad i \in [B]$

```
/* Gradient non-tilted inner product */
```

Compute $V_{\phi,\psi}^i = \nabla_{\phi,\psi} k_{\phi,\psi}(x_i, z_i) = \nabla_{\phi,\psi}\left(f_\phi(x_i)^\top s_\psi(z_i)\right)$

```
/* Compute combined gradient */
```

Return $\hat{\nabla}_{\phi,\psi} = -\frac{1}{B} \sum_{i=1}^B \left(V_{\phi,\psi}^i - U_{\phi,\psi}^i\right)$

---

where $q_{\phi,\psi}$ denotes the exponentially tilted density constructed in Section 3.2. Accordingly, the form of gradient updates for the LBF-NPE procedure is extremely simple; in fact, holding $\psi$ constant, the gradient with respect to $\phi$ is that of an affine function of the network outputs $f_\phi$. The same relationship holds when taking gradients for $\psi$ holding $\phi$ constant. Targeting such simple functions for optimization, besides being simple to implement, benefits from a convex formulation (see Section 4.2 below). The invariance of the inner product $k_{\psi,\phi}$ under arbitrary rescalings of $f$ and $s$, on the other hand, complicates optimization: this motivates a variant of LBF-NPE that reparameterizes outputs to unit norm (see Section 4.4).

## 4.2 CONVEXITY

McNamara et al. (2024a) show that neural posterior estimation (NPE) optimizes a convex functional objective function provided that the variational family is log-concave in $f$, the inference network. This ensures the forward KL objective of NPE (cf. Equation 3) is convex in $f$. Recent advances in the study of wide networks via the neural tangent kernel (NTK) (Jacot et al., 2018) have shown that fitting network parameters to minimize convex loss functionals (e.g., mean squared error) follows kernel gradient descent dynamics to a global optimum in the infinite-width limit (Jacot et al., 2018; McNamara et al., 2024a).

The amortized forward KL objective function that we target (cf. Equation 6) benefits from these same properties. For an arbitrary collection of basis functions, the objective function $L_{\text{LBF-NPE}}$ remains a convex functional in $f$. Likewise, for fixed $f$, $L_{\text{LBF-NPE}}$ is a convex functional in $s$. We formalize this in the proposition below.

**Proposition 1.** *The functional*

$$L(f, s) = -\mathbb{E}_{p(z,x)}\left(f(x)^\top s(z) - \log\left(\int \exp\left(f(x)^\top s(\tilde{z})\right) h(\tilde{z}) d\tilde{z}\right)\right)$$

*is marginally convex in the arguments $f$ and $s$, respectively.*

A proof and additional discussion are provided in Appendix B. Proposition 1 shows that in the case where either $f$ or $s$ is fixed, the resulting objective function is fully convex in the remaining argument. This observation motivates a variant of LBF-NPE where the basis functions are fixed a priori (see Section 4.3).

## 4.3 FIXED BASIS FUNCTIONS

Rather than adaptively fitting basis functions $s_\psi$, the practitioner may simply use a fixed basis defined ahead of time. This approach is motivated by the convexity of the resulting functional in $f$, as well as the approaches of related work based on basis expansion parameterizations, which use fixed orthonormal eigenfunctions (cf. Section 5). In this variant of LBF-NPE, the objective function $L(\phi, \psi)$ collapses to the marginal $L(\phi)$ for optimization. As we elaborate in Section 4.2, LBF-NPE

has a convex formulation in this setting, which empirically results in advantageous optimization trajectories relative to competing methods (we demonstrate this in Section 6.1).

Several choices of basis may be of interest to practitioners. EigenVI, a related basis-expansion method for VI (cf. Section 5), utilizes a (truncated) orthonormal basis of eigenfunctions, such as Bernstein, Legendre, or Hermite polynomials (Cai et al., 2024). Selecting a large $K$ improves faithfulness to the complete basis, but doing so increases the dimension of the optimization problem, exponentially so in multiple dimensions. Further, as generally such basis functions are *global* (i.e., nonzero on all of the latent space $I$), in this design *every* basis function contributes to the density value $q(z)$ at every point $z$; this may make it difficult to control the local behavior of the fitted density.

An alternative approach is to model $\log q(z; \eta) = \log q(z; f_\phi(x))$ via a *local* basis expansion. We specialize to B-splines (Appendix A.1) and wavelets (Appendix A.2) in our experiments, two rich families that we recommend for practitioners. In this framework, each basis function is nonzero only in a small neighborhood of the latent support. Locality of the basis functions simplifies optimization by inducing a sparser problem than a set of *global* basis functions would. For a single Monte Carlo draw $(z^*, x^*) \sim p(z, x)$, the gradient $-\nabla_\eta \log q(z^*; \eta)\mid_{\eta = f_\phi(x^*)}$ is nonzero at only a few indices because many basis functions are zero at any given $z^*$.

### 4.4 Reducing Degeneracy through Stereographic Projection

As noted in Section 4.1, both gradients and the log-density itself only depend on the inner product $f_\phi(x)^\top s_\psi(z)$. Adaptively learning both the inference network $f_\phi$ and basis function network $s_\psi$ thus suffers from an inherent lack of identifiability: different rescalings or rotations of the vectors defined by $f_\phi$ and $s_\psi$ can lead to the same loss function values, since the loss function (Equation 6) only depends on the inner product. To mitigate this degeneracy, we propose a variant of LBF-NPE that uses stereographic projection reparameterization to normalize the output tensor onto the unit hypersphere. This resolves the rescaling degeneracy, but rotational degeneracy persists even after the projection. Specifically, for a $K$-dimensional coefficient or basis function vector, we construct a neural network that outputs a $(K-1)$-dimensional vector $u$. We then apply the stereographic projection reparameterization: $y = \left( \frac{2u}{1+\|u\|^2}, \frac{1-\|u\|^2}{1+\|u\|^2} \right)$

which maps $u \in \mathbb{R}^{K-1}$ to a unit vector $y \in \mathbb{R}^K$ such that $\|y\| = 1$. This normalization mitigates identifiability issues, and the reparameterization yields strong results in our experiments (see Appendix D for additional discussion). Our loss function takes the following form when we apply this reparameterization:

$$\hat{L}_{\text{LBF-NPE}}(\phi, \psi) = \mathbb{E}_{p(z,x)} \left( -w \hat{f}_\phi(x)^\top \hat{s}_\psi(z) + \log \left( \int \exp \left( w \hat{f}_\phi(x)^\top \hat{s}_\psi(\tilde{z}) \right) h(\tilde{z}) d\tilde{z} \right) \right), \quad (12)$$

where $\hat{f}_\phi(\cdot)$ and $\hat{s}_\psi(\cdot)$ are reparameterized network outputs, and $w$ is a fixed scaling hyperparameter.

## 5 Related Work

Exponential family distributions are a common class of distributions for both traditional variational inference and NPE. In the simplest cases, Gaussian, Bernoulli, and other "simple" exponential families are used (Liu et al., 2023; Cranmer et al., 2020; Blei et al., 2017). Typically, however, these distributions are not parameterized in canonical form (where $\eta = f_\phi(x)$ is the natural parameter of the family). However, for NPE, McNamara et al. (2024a) recommend using the canonical parameterization, even for simple families such as the Gaussian, to benefit from convex loss (Section 4.2). General exponential families parameterized by neural networks were first proposed in Pacchiardi & Dutta (2022) to represent the *likelihood* function for likelihood-free settings. Akin to Approximate Bayesian Computation (ABC) methods, this approach aims to learn low-dimensional summary statistics of $x$ to represent the likelihood, and subsequently performs inference with potentially expensive MCMC or ABC routines. We compare to this approach in Appendix E.6. To our knowledge, LBF-NPE is the first method to utilize neural exponential families to represent the posterior distribution and to use this family within the amortized inference setting. LBF-NPE is also unique in exploiting the low dimensionality of the posterior projections of interest.

Parameterizing variational distributions via basis expansions is a relatively new line of research; a recent non-amortized approach, EigenVI (Cai et al., 2024), presents an algorithm for optimizing a score-based divergence with a variational family parameterized via a linear combination of orthogonal eigenfunctions. Key limitations of this approach are i) the lack of amortization and ii) the necessity of utilizing orthogonal, fixed eigenfunctions as the basis: truncation of these bases necessarily introduces approximation error. In our case, $s_\psi$ is unrestricted: the basis functions can be arbitrary, and so a fixed number $K$ may be sufficient for some classes of posteriors (cf. Section 6.2).

Mixtures of Gaussians and normalizing flows are other common choices of variational families for NPE (Gershman et al., 2012; Papamakarios & Murray, 2016; Papamakarios et al., 2021; Rezende & Mohamed, 2015). Although more flexible than simple exponential families, these parameterizations may suffer from convergence to shallow local optima during optimization (cf. Section 6.1). We compare LBF-NPE to mixtures of Gaussians, normalizing flows, as well as EigenVI in our experiments.

## 6    EXPERIMENTS

In numerical experiments, we fit a variety of complex posterior distributions using LBF-NPE. In Sections 6.1 and 6.4, we infer one-dimensional posterior projections using the variant of our method with fixed basis functions, whereas in Sections 6.2 and 6.3, we infer two-dimensional posterior projections using adaptive basis functions. Additional details about each of these experiments appear in Appendix D.

NPE with various alternative variational families serves as our primary benchmarks; we can compare to these methods quantitatively by assessing the NPE objective with each choice of variational distribution. We benchmark NPE with variational families based on mixture density networks (MDNs), RealNVP, and neural spline flows (Papamakarios & Murray, 2016; Durkan et al., 2019). In addition to the results in this section, additional results appear in Appendix E, including results from comparisons to two non-NPE-based variational inference methods: EigenVI (Appendix E.5) and a score-matching neural-likelihood-based method for likelihood-free inference (Appendix E.6).

### 6.1    TOY EXAMPLE: SINUSOIDAL LIKELIHOOD

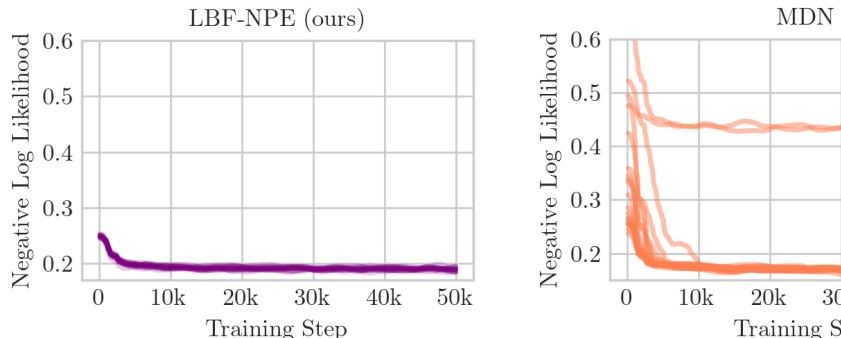

Figure 1: Negative log likelihood of our method and the MDN. Each model is trained with 20 different random seeds, and the records are smoothed using a Gaussian filter with $\sigma = 20.0$.

We first exhibit the advantages of LBF-NPE's convex variational objective by demonstrating consistent convergence on a highly multimodal problem with fixed basis functions (cf. Section 4.3). The model draws an angle $z \sim \text{Unif}[0, 2\pi]$ followed by $x \mid z \sim \mathcal{N}\left(\sin(2z), \sigma^2\right)$ for fixed $\sigma^2 = 1$. The exact posteriors $p(z \mid x)$ have up to four modes, depending on the realization $x$. We compare LBF-NPE, using a fixed collection of 14 B-spline functions of degree two on a mesh of $[0, 2\pi]$ (see Appendix A.1 for additional detail on B-splines), and a mixture density network (MDN) (Bishop, 1994; Papamakarios & Murray, 2016), using a mixture of five Gaussian distributions. Both variational distributions have 14 distributional parameters per observation $x$ (there are four mixture weight parameters for the MDN due to the simplex constraint). Additional experimental details are given in Appendix D. Figure 1 shows that for 20 different runs of the optimization routine, LBF-NPE

Table 1: Forward KL divergence (FKL), reverse KL divergence (RKL), and negative log-likelihood (NLL) of LBF-NPE (ours), NSF (Neural Spline Flow), RealNVP, and MDN on three 2D test cases. Lower values indicate better posterior approximation.

| | **Forward KL Divergence** | | | |
| --- | --- | --- | --- | --- |
| | **LBF-NPE** | **NSF** | **RealNVP** | **MDN** |
| **Bands** | **0.0048** (± **0.0003**) | 0.016 (± 0.003) | 0.015 (± 0.005) | 0.182 (± 0.01) |
| **Ring** | **0.0054** (± **0.0005**) | 0.017 (± 0.004) | 0.024 (± 0.005) | 0.205 (± 0.02) |
| **Spiral** | **0.187** (± **0.004**) | 0.201 (± 0.01) | 0.545 (± 0.07) | 0.948 (± 0.09) |

| | **Reverse KL Divergence** | | | |
| --- | --- | --- | --- | --- |
| | **LBF-NPE** | **NSF** | **RealNVP** | **MDN** |
| **Bands** | **0.0014** (± **0.0004**) | 0.0099 (± 0.001) | 0.011 (± 0.007) | 0.156 (± 0.02) |
| **Ring** | **0.0027** (± **0.0003**) | 0.013 (± 0.003) | 0.014 (± 0.003) | 0.204 (± 0.01) |
| **Spiral** | **0.188** (± **0.005**) | 0.322 (± 0.04) | 0.666 (± 0.09) | 1.973 (± 0.14) |

| | **Negative Log-Likelihood** | | | |
| --- | --- | --- | --- | --- |
| | **LBF-NPE** | **NSF** | **RealNVP** | **MDN** |
| **Bands** | **-0.060** (± **0.07**) | 0.151 (± 0.23) | 0.157 (± 0.22) | 1.389 (± 0.41) |
| **Ring** | **0.030** (± **0.03**) | 0.621 (± 0.24) | 0.733 (± 0.11) | 1.031 (± 0.18) |
| **Spiral** | 0.838 (± 0.13) | **0.727** (± **0.25**) | 0.859 (± 0.32) | 2.788 (± 0.31) |

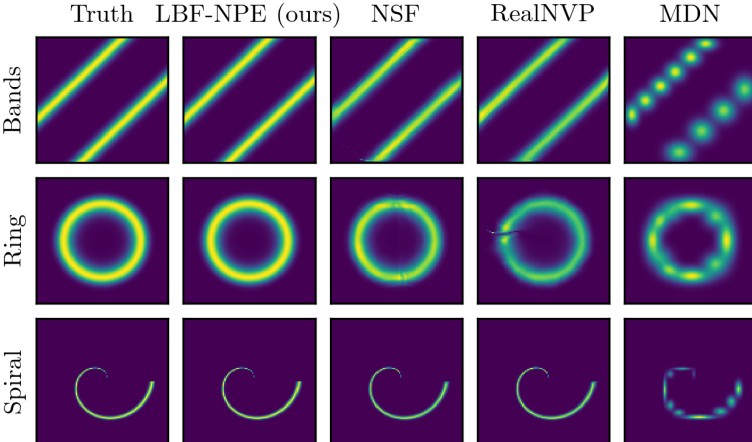

Figure 2: Example posteriors of three problems in two dimensions. NSF refers to Neural Spline Flow.

consistently converges to the same solution, whereas the MDN sometimes converges to a suboptimal local optimum. Visualizations of posterior approximations from both methods are provided in Appendix E.

## 6.2 COMPLEX MULTIVARIATE REPRESENTATIONS IN 2D

We showcase LBF-NPE on three test problems in two dimensions, named "banded", "ring", and "spiral," and visualized in the left column of Figure 2. Each model consists of a two-dimensional latent variable $z \in \mathbb{R}^2$ and an observation $x \in \mathbb{R}$, and in some cases nuisance latent variables as well. Further details of the generative processes for these three problems are provided in Appendix D (Sections D.2 to D.4).

LBF-NPE is able to approximate these complex posteriors nearly perfectly using only 20 adaptive basis functions $s_\psi$. Both the amortization function $f_\phi$ and the basis functions $s_\psi$ are parameterized

using deep neural networks. For additional implementation details, we refer the reader to Appendix D. We follow Algorithm 1 and evaluate both the variational posterior $q(z; f_\phi(x), s_\psi(z))$ and the exact posterior $p(z \mid x)$ on a fine mesh grid that covers the support of the posterior. Qualitative results appear in Figure 2 and quantitative results appear in Table 1 for held-out test points. LBF-NPE outperforms the MDN and multiple types of normalizing flow in nearly all cases. We provide additional visualizations of the variational approximations found by LBF-NPE and its competitors in Appendix E.

## 6.3 OBJECT DETECTION

We apply LBF-NPE with adaptive basis functions to the problem of object detection in astronomical images. We use a generative model resembling the scientific model of Liu et al. (2023). In brief, this generative process first independently samples star locations $l_1, l_2 \sim \mathrm{Unif}([0, 16] \times [0, 16])$ and star fluxes $f_1, f_2 \sim \mathcal{N}(\mu, \sigma^2)$ for two objects. Afterward, a latent noise-free image $I$ is rendered by convolving these point sources with a point-spread function (PSF). Finally, given $I$, the intensity of each pixel $(j, k)$, for $j, k \in \{1, \ldots, 16\}$ is independently drawn as $x_{jk} \sim \mathrm{Poisson}(I_{jk})$, reflecting Poisson shot noise.

Figure 3 shows two examples of the noisy observations $x$, along with the posterior approximations for the locations of each. The posterior distribution for this problem is multimodal with a high degree of separation between modes. LBF-NPE parameterizes this shape effectively. Inference on locations of objects would initially appear to be difficult for LBF-NPE, as our method does not directly parameterize location parameters: instead, the learned basis functions must be expressive enough to parameter-

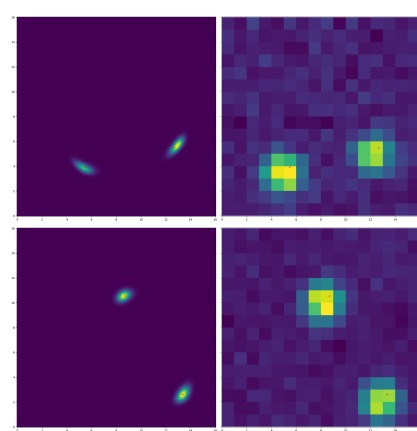

Figure 3: Two example posteriors (left) conditional on the observed images $x$ (right). In each case, LBF-NPE correctly recovers the locations of the two objects.

ize arbitrary pairs of separated modes. In our ablations, we vary the number of basis functions $K = 9, 20, 36, 64$ for LBF-NPE and provide visualizations in Appendix E.3. Examining the form of the fitted basis functions across varying $K$ for this problem is particularly illustrative of the expressivity of the adaptive approach to fitting the basis functions compared to a fixed basis function framework.

## 6.4 CASE STUDY: REDSHIFT ESTIMATION

The redshift of galaxies is a key quantity of interest as it characterizes their distances from Earth. Redshift measures the extent to which electromagnetic waves are "stretched" to redder wavelengths as objects move away from Earth. The distribution of redshifts across many objects is a powerful

Table 2: Total held-out NLL of the true redshift $z$.

|  | LBF-NPE | NSF | MDN |
|---|---|---|---|
| NLL | -57,220 (± 152) | -55,389 (± 379) | -50,648 (± 322) |

probe of cosmology (Malz & Hogg, 2022). Redshift estimation from photometric data (images) is referred to as photo-$z$ estimation. We extend the methodology of the Bayesian Light Source Separator (BLISS) package (Liu et al., 2023), a state-of-the-art package for probabilistic object detection in astronomical images, for this task, by adding a redshift density estimation "head" to the existing BLISS network. To each detected object, we associate a redshift probability density function fitted by LBF-NPE with a fixed B-spline basis family. LSST DESC DC2 Simulated Sky Survey dataset (LSST Dark Energy Science Collaboration et al., 2021) serves as the generative model, providing simulated $(z, x)$ pairs, where $z$ denotes redshift and $x$ are the astronomical images. This highly realistic dataset consists of mock catalogs of astronomical images generated by directly modeling known physical quantities of the universe using empirical priors and physics-informed modeling. Appendix D provides further details of the experimental setting.

We compare LBF-NPE with variational families based on neural spline flow (NSF) and a mixture density network (MDN), all embedded with the BLISS framework. The MDN uses five Gaussian components, in keeping with state-of-the-art work on photo-$z$ estimation (Merz et al., 2025); adding more components did not improve the quality of fit. The only difference between the two approaches is the choice of parameterization for the variational family. We compute the negative log-likelihood (NLL) of a held-out test set of 153,000 astronomical objects. Table 2 shows that LBF-NPE with the B-spline variational family parameterization outperforms both the MDN and the NSF.

# 7 DISCUSSION & LIMITATIONS

LBF-NPE models the log density of the variational distribution as a linear combination of expressive basis functions, which is beneficial for several reasons. First, log-space modeling results in a *multiplicative* influence of different basis functions. Regions of latent space can effectively be "zeroed-out" more easily in this context compared to performing the modeling in density space directly. Second, our model of the log density results in an unconstrained optimization problem in $f_\phi$ and $s_\psi$: the coefficients and basis functions may be either positive or negative, whereas other density estimation methods may require nonnegativity or other constraints on the coefficients or basis functions to obtain a valid density that integrates to unity (Cai et al., 2024; Koo & Kim, 1996; Kirkby et al., 2023). Third, unlike other methods, LBF-NPE may be applied to the growing class of "likelihood-free inference" problems (Cranmer et al., 2020; Thomas et al., 2022) as fitting the variational posteriors only requires samples from the joint model $p(z, x)$. The advantageous marginalization properties of the forward KL criterion pair nicely with this setting: the generative model can have arbitrary nuisance latent variables that are implicitly marginalized out (Ambrogioni et al., 2019). We provide an additional example marginalizing over the parameters of a Bayesian neural network (BNN) in Appendix E.8.

Using basis expansions to parameterize variational distributions is a recent and exciting innovation in variational inference. Relative to EigenVI, LBF-NPE performs better with fewer basis functions (Appendix E.5). This may be due to removing orthogonality constraints and adaptively fitting basis functions: by averaging across posteriors for arbitrary $x$, our approach implicitly regularizes the basis functions, preventing overfitting to any single instance. Appendix E contains additional discussion and visualization of fitted basis functions $s_\psi$.

The main limitation of LBF-NPE is the difficulty of sampling from the variational distribution. LBF-NPE directly fits the log density of the variational distribution, but samples from this density are typically needed for inference. The low dimensionality of many NPE targets ensures that sampling is straightforward: inverse transform sampling is readily applicable (cf. Appendix C). For higher-dimensional targets, importance sampling may be adequate to estimate functionals with respect to the variational distribution.

Despite this sampling challenge, our approach demonstrates that basis expansion methods offer a compelling middle ground between optimization simplicity and expressivity for NPE. Future work could explore new bilevel optimization approaches (Xiao & Chen, 2025) to jointly learn adaptive basis functions alongside their coefficients. Additionally, extending our approach beyond low-dimensional targets to high-dimensional targets with simplifying structure—such as known or assumed conditional independencies—could broaden the applicability of basis expansion methods.

## ACKNOWLEDGMENTS

We thank the reviewers for their helpful comments and suggestions. This material is based on work supported by the National Science Foundation under Grant Nos. 2209720 (OAC) and 2241144 (DGE), and the U.S. Department of Energy, Office of Science, Office of High Energy Physics under Award Number DE-SC0023714.

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

## A  B-SPLINE & WAVELET BASIS FUNCTIONS

We give examples of two classes of local basis functions defined on the unit interval $[0, 1]$: B-splines and wavelets. Without loss of generality, these definitions can be extended to construct the family on any interval $I = [a, b]$ for $a, b \in \mathbb{R}$. These families are potential candidates for a local basis parameterization of the variational posterior.

### A.1  B-SPLINES

For a choice of degree $d \geq 1$ and a uniformly spaced set of points $\{t_i\}_{i=1}^K$, the B-splines are a set of local basis functions $\{b_i^{(d)}(z)\}_{i=1}^K$ that are defined recursively as follows (d. Boor, 1978; Eilers & Marx, 1996):

$$b_i^{(d)}(z) = 1 \text{ on } [t_i, t_{i+1}) \text{ o.w. } 0 \text{ for all } i, z, \text{ if } d = 0 \tag{13}$$

$$\frac{b_i^{(d)}(z)}{(t_{i+d} - t_i)} = \frac{z - t_i}{t_{i+d-1} - t_i} \cdot \frac{b_i^{(d-1)}(z)}{t_{i+d-1} - t_i} + \frac{t_i - z}{t_{i+d} - t_i} \cdot \frac{b_{i+1}^{(d-1)}(z)}{t_{i+d} - t_i} \quad d \geq 1. \tag{14}$$

The B-spline basis functions $b_i^{(d)}(z)$ are thus individually piecewise polynomial splines of degree $d$. While each function $b_i^{(d)}$ is symmetric about the $i$th knot $t_i$ (or a midpoint of two knots), it is nonzero for a range of only $2d$ knots, in accordance with the *locality* aspect described above (see Figure 4).

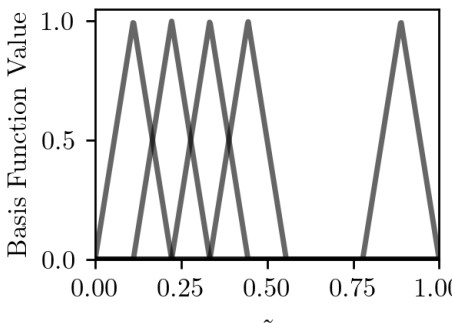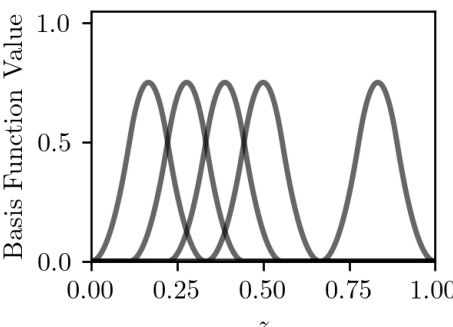

Figure 4: Example visualizations of B-Spline basis functions of degree 1 (left) and 2 (right). For brevity, we show only a subset of the basis functions.

### A.2  WAVELETS

A collection of wavelet local basis functions on $[0, 1]$ is defined relative to a "mother wavelet" function denoted $H$. For ease of exposition, we consider the Haar wavelet (Koo & Kim, 1996; Steele, 2010) given by

$$H(z) = \begin{cases} 1, 0 \leq z < \frac{1}{2} \\ -1, \frac{1}{2} \leq z \leq 1 \\ 0, \text{ otherwise} \end{cases} \tag{15}$$

Thereafter, the set of local basis functions $b_i$ is defined recursively as follows: writing each $i$ uniquely as $i = 2^j + k, j \geq 0, 0 \leq k < 2^j$, we have

$$b_i(z) = 2^{j/2} H(2^j \cdot z - k). \tag{16}$$

The local basis functions are thus defined as shifted and scaled versions of the mother wavelet $H$. One can check that $b_i(z)$ is nonzero only on the interval $[k \cdot 2^{-j}, (k+1) \cdot 2^{-j}]$ for $i = 2^j + k$, so the basis functions become nonzero on increasingly local regions even for moderate values of $i$ (say, $i = 200$). Wavelets are often described as being local with respect to both space *and* frequency as $b_i$ becomes increasingly "spiky" as well due to the coefficient $2^{j/2}$ (Steele, 2010).

# B    CONVEXITY

## B.1    CONVEXITY & CONVERGENCE OF LBF-NPE WITH FIXED BASIS FUNCTIONS

As referenced in Section 4.2, in the setting where the basis functions $s_\psi$ are fixed ahead of time, the objective function of LBF-NPE becomes a convex functional of the amortization network $f$. In this case, our setting can be shown to be globally convergent under suitable regularity conditions on the network architecture, in the asymptotic limit as the network width tends arbitrarily large. We restate this result, proven in McNamara et al. (2024a), below.

**Proposition.** *Let $\mathcal{X} \subseteq \mathbb{R}^d$ and $\mathcal{Y} \subseteq \mathbb{R}^K$. Let $f_\phi = f(\cdot; \phi) : \mathcal{X} \to \mathbb{R}^K$ be parameterized as a scaled two-layer ReLU network of width $p$, i.e. $f_i(x; \phi) = \frac{1}{\sqrt{p}} \sum_{j=1}^p a_{ij} \sigma(x^\top w_j)$ for $i = 1, \ldots, K$. Define the loss functional*

$$\ell(x, \eta) = \mathrm{KL}(p(z \mid x) \,\|\, q(z; \eta)),$$

*and allow parameters $\phi = \{a_{ij}, w_j\}, i = 1, \ldots, K, j = 1, \ldots p$ to evolve via the gradient flow ODE $\dot{\phi}(t) = -\nabla_\phi \mathbb{E}_{p(x)} \ell(x, f(x; \phi(t)))$. Then we have the following results (under regularity conditions (A1)–(A6)):*

- *a) $L_{\text{LBF-NPE}}(\phi(t))$ is precisely $\mathbb{E}_{p(x)} \ell(x, f(x; \phi(t)))$, optimized by the gradient flow above. Further, the functional $M_{\text{LBF-NPE}}(f) : f \mapsto \mathbb{E}_{p(x)} \ell(x, f(x))$ is a convex functional in $f$, with a global optimum $f^*$.*

- *b) For the parameterization above, with $\phi$ following the gradient flow ODE, we have that there exists $T > 0$ such that*

$$\lim_{p \to \infty} M_{\text{LBF-NPE}}(f_T) \le M_{\text{LBF-NPE}}(f^*) + \epsilon,$$

*where $f_T = f(\cdot; \phi(T))$.*

The proposition above, proven in (McNamara et al., 2024a), states that gradient descent on the convex functional $\mathbb{E}_{p(x)} \ell(x, f(x; \phi))$ converges arbitrarily close to its optimum in the infinite-width limit, relying on universality results of shallow networks and NTK theory (Jacot et al., 2018; Lee et al., 2022). Our parameterization of the variational distribution $q$ as an exponential family in LBF-NPE falls into this setting when the basis functions are fixed, allowing us to directly apply results from (McNamara et al., 2024a). We refer to the proofs therein rather than a restatement here.

Regularity conditions sufficient for the above to hold are provided below. They assume a well-behaved functional $M$, a particular initialization of the width-$p$ network, and uniform boundedness conditions on gradients along the optimization trajectory. Although we emphasize that NTK-style results only approximately explain the success of our method or other neural posterior estimation methods in practice (the infinite-width limit can only be approximated by finite networks, and the continuous gradient flow ODE is approximated by stochastic gradient descent), results like these prove that LBF-NPE benefits from an advantageous optimization landscape asymptotically.

- (A1)  The data space $\mathcal{X}$ is compact.

- (A2)  Weights are initialized as $a_{ij} \overset{a.s.}{=} 0$, $w_j \overset{iid}{\sim} \mathcal{N}(0, I_d)$.

- (A3)  The neural tangent kernel at initialization, $K_\phi(x, x') = Jf(x; \phi) Jf(x'; \phi)^\top$ at $\phi = \phi(0)$, is dominated by some integrable random variable $G$, uniformly over $x, x'$.

- (A4)  The gradient $\nabla_\eta \ell(x, \eta)$ is uniformly bounded for all $x, \eta$.

- (A5)  The limiting NTK $K_\infty = \lim_{p \to \infty} K_\phi$ is positive-definite (we note that the existence of the limit is guaranteed).

- (A6)  The functional $M_{\text{LBF-NPE}}$ is bounded below; and its minimizer $f^*$ has finite norm with respect to the RKHS norm of the limiting NTK $K_\infty$.

- (A7)  The function $\ell(x, \eta)$ is $C$-smooth in $\eta$ for some $C < \infty$.

## B.2 CONVEXITY OF LBF-NPE

In this subsection, we prove the convexity of the joint functional $L(f, s)$ in Proposition 1. Neural network theory and NTK-based analysis of this objective in $s$ are beyond the scope of this work. We prove marginal convexity in $s$, and appeal to previous NTK-based results (such as the above), which motivate our empirical results: optimization of convex functionals of neural network outputs is advantageous compared to the optimization of nonconvex functionals due to the preferable optimization landscape of the former (Bach, 2017; Bengio et al., 2005; Jacot et al., 2018; Wojtowytsch, 2020).

We first present Hölder's inequality, which we'll use in the proof.

**Lemma 1** (Hölder). *If $S$ is a measurable subset of $\mathbb{R}^n$ (with respect to Lebesgue measure), and $f$ and $g$ are measurable real-valued functions on $S$, then Hölder's inequality is*

$$\int_S |f(x)g(x)|\mathrm{d}x \leq \left(\int_S |f(x)|^p\,\mathrm{d}x\right)^{\frac{1}{p}}\left(\int_S |g(x)|^q\,\mathrm{d}x\right)^{\frac{1}{q}}.$$

*for any $p, q$ satisfying $\frac{1}{p} + \frac{1}{q} = 1$.*

Hölder's inequality will be used to prove Proposition 1, restated below.

**Proposition 1.** *The functional*

$$L(f, s) = -\mathbb{E}_{p(z,x)}\left(f(x)^\top s(z) - \log\left(\int \exp\left(f(x)^\top s(\tilde{z})\right)h(\tilde{z})d\tilde{z}\right)\right)$$

*is marginally convex in the arguments $f$ and $s$, respectively.*

*Proof.* Ignore the outer expectation for now, and consider a fixed $x, z$ drawn from $p(z, x)$. The inner product $-f(x)^\top s(z)$ is clearly convex in $s$. We now turn to the more complicated expression, $\log\left(\int \exp\left(f(x)^\top s(z')\right)dz'\right)$. Note that the integral is over $z'$ this expression and doesn't depend on the realization $z$. It still depends on $s$, however.

We prove this function is convex as follows. Let $\alpha \in (0, 1)$, and consider functions $s_1, s_2$. Then

$$\log\left(\int \exp\left(f(x)^\top [\alpha s_1(z') + (1-\alpha)s_2(z')]\right)dh(z')\right)$$

$$= \log\left(\int \exp\left(\alpha f(x)^\top s_1(z')\right)\exp\left((1-\alpha)f(x)^\top s_2(z')\right)dh(z')\right)$$

$$= \log\left(\int \left[\exp\left(f(x)^\top s_1(z')\right)\right]^\alpha \left[\exp\left(f(x)^\top s_2(z')\right)\right]^{1-\alpha}dh(z')\right)$$

$$= \log\left(\int [u(z')]^\alpha [v(z')]^{1-\alpha}dh(z')\right)$$

where we have defined $u(z') = \exp(f(x)^\top s_1(z'))$ and $v(z') = \exp(f(x)^\top s_2(z'))$. Consider the integral above, and apply Hölder's inequality with $1/p = \alpha$ and $1/q = 1 - \alpha$. These sum to one as required. We take $f$ to be $[u(z')]^\alpha$ and $g$ to be $[v(z')]^{(1-\alpha)}$.

Continuing, we have

$$= \log \left( \int [u(z')]^{\alpha} \cdot [v(z')]^{(1-\alpha)} \, dh(z') \right) \quad \text{[this line repeated for clarity]}$$

$$\leq \log \left( \left[ \int ([u(z')]^{\alpha})^{1/\alpha} \, dh(z') \right]^{\alpha} \cdot \left[ \int \left( [v(z')]^{(1-\alpha)} \right)^{1/(1-\alpha)} \, dh(z') \right]^{1-\alpha} \right) \quad \text{[Hölder]}$$

$$= \alpha \log \left( \int u(z') dh(z') \right) + (1-\alpha) \log \left( \int v(z') dh(z') \right)$$

$$= \alpha \log \left( \int \exp(f(x)^{\top} s_1(z')) dh(z') \right) + (1-\alpha) \log \left( \int \exp(f(x)^{\top} s_2(z')) dh(z') \right).$$

This was all that is required to show that the mapping $s \mapsto \log \left( \int \exp \left( f(x)^{\top} s(z') \right) dz' \right)$ is convex. As the sum of two convex functions is convex, we've shown convexity of the integrand for any fixed draw $z, x \sim p(z, x)$. To conclude, we observe that by linearity of integration, this holds for the integral as well. The argument for $f$ is identical by symmetry of the inner product.

$\square$

## C  SAMPLING

Similar to EigenVI (Cai et al., 2024), LBF-NPE does not easily admit sampling from the fitted variational density. This is one limitation of the nonparametric nature of the density model in both of these approaches to variational inference.

In the low-dimensional case, sampling can be performed by inverse transform sampling. In this approach, one uses the cumulative distribution function $Q$ of $q$, defined as

$$Q(z^*) = P_q(z < z^*) = \int_{-\infty}^{z^*} q(z) dz$$

where $q(z)$ is the fitted variational density (say, conditional on some datum $x$ of interest). The function $Q$ is invertible. Sampling can be performed by drawing $U \sim \text{Unif}[0, 1]$, and thereafter computing

$$Z = Q^{-1}(U).$$

The result of this procedure is a draw $Z \sim q(z)$. Computing and inverting cumulative distribution functions in low dimensions is fairly straightforward. As outlined in the main text, this low-dimensional setting is one we commonly find to be of use to practitioners, especially for the types of problems we consider in our experiments.

To sample from high-dimensional posteriors, several different approaches are available. One approach, as outlined in (Cai et al., 2024), is sequential sampling, whereby one samples

$$q(z_1), q(z_2 \mid z_1), q(z_3 \mid z_1, z_2), \ldots, q(z_d \mid z_1, \ldots, z_{d-1})$$

in order. Each individual density above can be sampled using the inverse transform sampling approach outlined above; conditioning within the exponential family parameterization is easily accomplished by freezing the already sampled indices and changing the variables of integration in the log integral. More generally, one could also utilize rejection sampling or other Monte Carlo sampling algorithms. As the unnormalized variational density has an extremely simple form, i.e., $q(z) \propto \exp(\eta^{\top} b(z))$, Markov chain Monte Carlo algorithms could also be an efficient way to sample from the distribution defined by the fitted density. We present some selected results of Langevin sampling and inverse transform sampling in Appendix C.1 to illustrate the utility of these approaches.

### C.1  RESULTS OF LANGEVIN DYNAMICS AND INVERSE TRANSFORM SAMPLING

We present the sampling results obtained via Langevin dynamics (Song et al., 2021) and inverse transform sampling for the three 2D case studies: bands, ring, and spiral. In addition to the inverse transform sampling described in the preceding section, we explore the use of Langevin

dynamics, a widely adopted method for generating samples in score-based generative models. This approach iteratively updates a set of particles according to:

$$dz_t = \epsilon \nabla_z \log p(z_t \mid x) dt + \sqrt{2\epsilon} dW_t,$$

where $\epsilon = 0.001$, $dW_t \sim \mathcal{N}(0,1)$, and $t \in \{1, 2, \ldots, 1000\}$. Since our model provides a differentiable approximation of $\log p(z_t \mid x)$ through $f_\phi(x)^\top s_\psi(z)$, the gradient $\nabla_z \log p(z_t \mid x)$ can be directly estimated. This enables us to apply Langevin dynamics for posterior sampling.

For each of the `bands`, `ring`, and `spiral` case studies, we generate 10,000 samples and visualize both the samples and their marginal densities in Figures 5 to 10. As illustrated in these figures, both Langevin dynamics and inverse transform sampling yield samples that closely match the estimated posterior distributions.

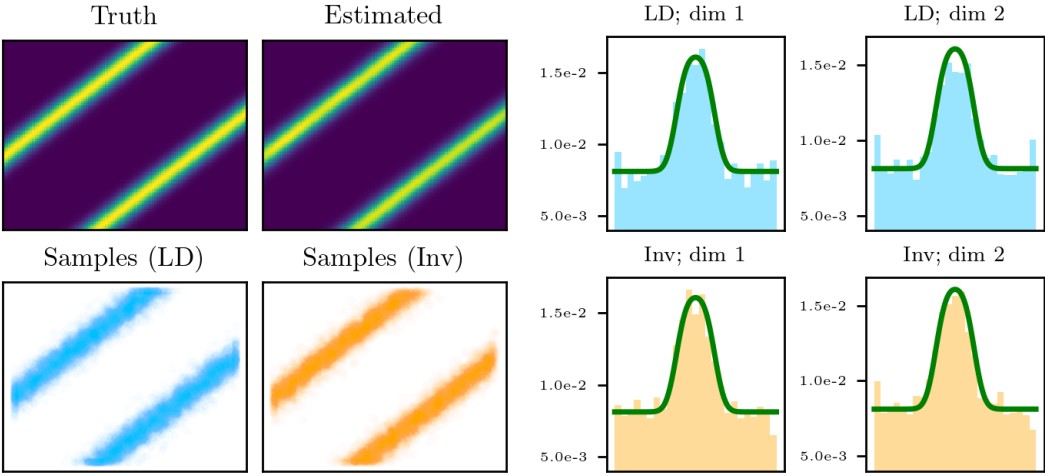

Figure 5: Sampling results for `bands`. "LD" and "Inv" denote Langevin dynamics and inverse transform sampling, respectively.

Figure 6: Marginal density of samples for `bands`. The green line indicates the estimated posterior density.

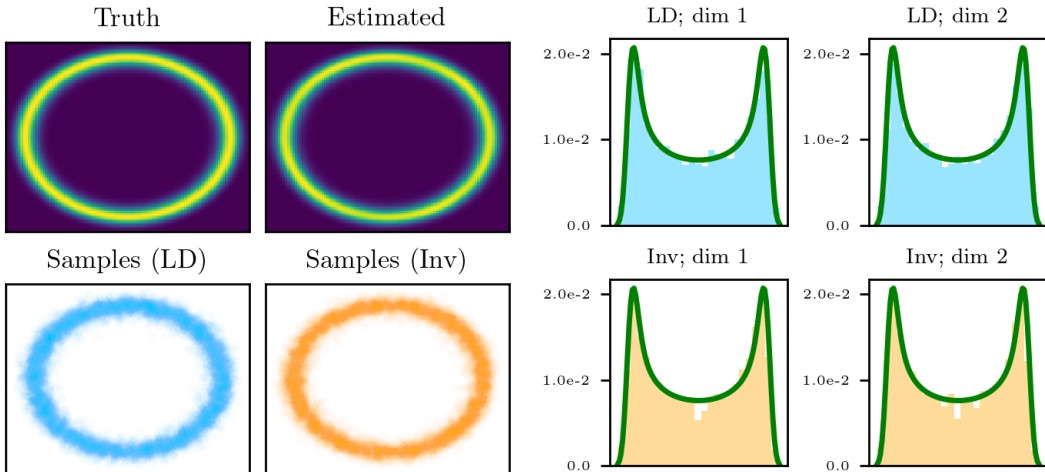

Figure 7: Sampling results for `ring`. "LD" and "Inv" denote Langevin dynamics and inverse transform sampling, respectively.

Figure 8: Marginal density of samples for `ring`. The green line indicates the estimated posterior density.

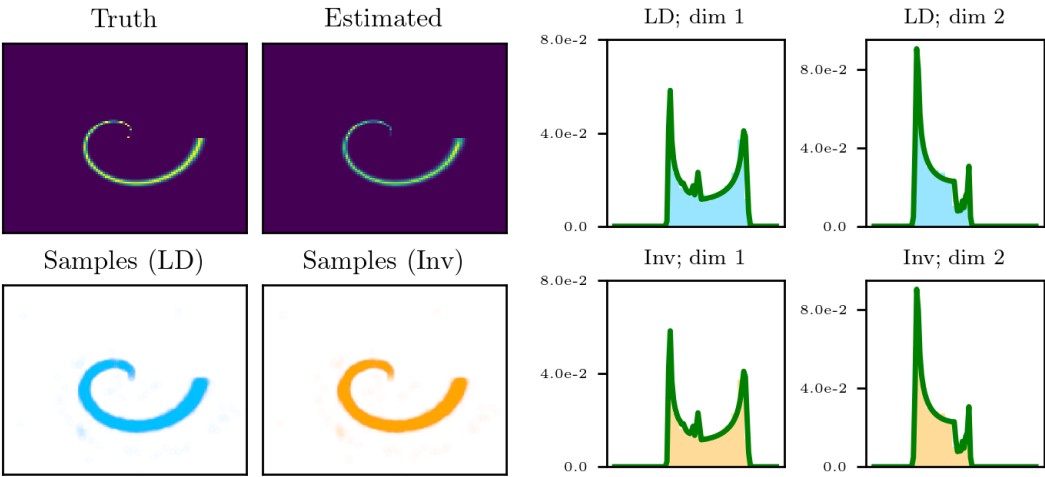

Figure 9: Sampling results for `spiral`. "LD" and "Inv" denote Langevin dynamics and inverse transform sampling, respectively.

Figure 10: Marginal density of samples for `spiral`. The green line indicates the estimated posterior density.

## D  EXPERIMENTAL DETAILS

Code to reproduce results is provided at `https://github.com/YicunDuanUMich/LBF-NPE`. We use PyTorch (Paszke et al., 2019; Ansel et al., 2024) and JAX (Bradbury et al., 2018). We also use the `equinox` library for deep learning in JAX (Kidger & Garcia, 2021). For Section 6.4, the DC2 Simulated Sky Survey data is publicly available from the LSST Dark Energy Science Collaboration (LSST DESC) (LSST Dark Energy Science Collaboration et al., 2021; 2022). All experiments are conducted on an Ubuntu 22.04 server equipped with an NVIDIA RTX 2080 Ti GPU.

### D.1  TOY EXAMPLE: SINUSOIDAL LIKELIHOOD

In Section 6.1, we design a simple Bayesian model to evaluate the convergence behavior of LBF-NPE and MDN. This statistical model is

$$z \sim \text{Unif}[0, 2\pi],$$
$$x \mid z \sim \mathcal{N}(\sin(2z), \sigma^2),$$

where $\sigma^2 = 1$. This model induces a multimodal posterior:

$$P(z \mid x) \propto \exp\left(-(\sin(2z) - x)^2 / (2\sigma^2)\right) \times \mathbb{I}(0 \le z \le 2\pi),$$

which exhibits two modes when $x \ge 1$ or $x \le -1$, and four modes otherwise.

For LBF-NPE, we construct a multilayer perceptron (MLP) to predict the coefficient vector $\eta = f_\phi(x)$, and the sufficient statistics $b(z) \in \mathbb{R}^K$ are computed using B-spline basis functions with $K = 14$. The MLP architecture consists of an input layer, four hidden layers, and an output layer, mapping $x \in \mathbb{R}$ to $\eta \in \mathbb{R}^K$. Each hidden layer includes a full connection layer of 128 units, layer normalization, and a ReLU activation. The output layer is linear. The B-spline basis comprises 14 degree-2 basis functions, with knots at $[0, 0, \text{linspace}(0, 2\pi, \text{num} = K), 2\pi, 2\pi]$. Although the B-spline basis $b(z)$ can be evaluated recursively as described in Section A.1, we precompute it on a grid to avoid redundant computation during training. Specifically, we pick 1000 uniformly spaced points in the interval $[0, 2\pi]$ and approximate the integral $\int \exp(\eta_i^\top b(z)) \, dz$ using the trapezoidal sum. For the term $f_\phi(x)^\top b(z)$, we use the basis vector corresponding to the grid point closest to the true latent variable $z_{\text{true}}$.

For MDN, we use the same MLP architecture, with the output adapted to represent the parameters of a mixture of 5 Gaussian distributions. The output vector has 10 parameters for means and variances, and 4 additional parameters for the mixture weights. The MDN objective is

$$L_{\text{MDN}}(\gamma) = -\mathbb{E}_{p(x,z)} \log q(z; t_\gamma(x)), \tag{17}$$

where $\gamma$ are the neural network parameters, $t_\gamma(x)$ denotes the predicted distribution parameters, and $q(z; \cdot)$ is the corresponding density.

The training procedures for LBF-NPE and MDN are identical, except for the loss function. At each step, we sample 1024 latent–observed pairs $(z, x)$ from the generative model and update model parameters using the AdaBelief optimizer (Zhuang et al., 2020) with a learning rate of 0.001. Training proceeds for 50,000 steps and completes within one hour for both methods. Peak GPU memory usage is approximately 8300MB. We hold out 1000 $(z, x)$ pairs and track their negative log-likelihood (NLL) over training, as shown in Figure 1. We apply Gaussian smoothing to the NLL curves with standard deviation $\sigma = 20$. This results in a smoothing kernel of size $161 = 4 \times 20 \times 2 + 1$, with weights given by $G_i = \exp(-i^2/(2\sigma^2))$ for $i \in \{-80, \ldots, 80\}$. With the normalization constants for $G_i$ omitted, the smoothed NLL at step $j$ is computed as

$$\text{NLL}_{\text{smooth},j} = \sum_{i=-80}^{80} \text{NLL}_{j+i} \cdot G_i.$$

### D.2 2D CASE STUDIES: BANDS

The statistical model for the `bands` test case, as introduced in Section 6.2, is

$$z_1, z_2 \sim \text{Unif}[-1, 1],$$
$$z = (z_1, z_2),$$
$$x \mid z \sim \mathcal{N}(|z_1 - z_2|, \sigma^2),$$

where $\sigma^2 = 10^{-2}$. The resulting posterior forms two elongated bands in the 2D latent space $P(z \mid x) \propto \exp\left(-(|z_1 - z_2| - x)^2/(2\sigma^2)\right) \cdot \mathbb{I}(-1 \leq z_1, z_2 \leq 1)$, with its maxima occurring along the lines where $|z_1 - z_2| = x$.

As the latent variable $z$ is now two-dimensional, LBF-NPE encounters increased complexity due to the larger number of basis functions required. In our LBF-NPE framework, both the coefficient network $f_\phi$ and the sufficient statistic network $s_\psi$ are implemented as multilayer perceptrons (MLPs) with four hidden layers, each containing 128 units. All layers use layer normalization to stabilize optimization and ReLU activations. The network $f_\phi$ maps input $x \in \mathbb{R}$ to a coefficient vector in $\mathbb{R}^K$, while $s_\psi$ maps $z \in \mathbb{R}^2$ to sufficient statistics in $\mathbb{R}^K$. We set $K = 20$ for consistency with other 2D case studies, though even $K = 2$ suffices to capture the posterior structure in this example (see Appendix E.3). The loss function for LBF-NPE follows Algorithm 1, where the integral term $\int \exp(\eta_i^\top s_\psi(z)) \, dz$ is approximated using a trapezoidal sum over a $100 \times 100$ uniform grid spanning $[-1, 1]^2$. During training, we alternate between updating $f_\phi$ and $s_\psi$: we train $f_\phi$ for 1000 steps while holding $s_\psi$ fixed, then train $s_\psi$ for 1000 steps with $f_\phi$ fixed, and repeat this process until the total training budget is exhausted. The choice of 1000 steps per phase is empirical; we observe diminishing returns in the loss reduction beyond 1000 steps, indicating that each sub-network has reached a near-optimal solution given the other is fixed. In addition, we use stereographic projection to reparameterize the output of $f_\phi$ and $s_\psi$.

For the MDN baseline, we use an MLP with the same architecture as $f_\phi$, except that it outputs a 50-dimensional vector representing the parameters of a mixture of 10 Gaussian components. Each component is parameterized by five values: two for the mean, two for the (diagonal) variance (assuming zero covariance), and one for the mixture weight. The loss function is identical to that described in Appendix D.1.

For the normalizing flow baseline, we adapt the classic coupling flow from (Dinh et al., 2017) to model the conditional posterior $p(z \mid x)$. Each coupling layer includes translation and scaling subnetworks conditioned on $x \in \mathbb{R}$. These sub-networks are implemented as MLPs, each taking as input the concatenation of the masked latent variable $z$ and the conditioning variable $x$. Each MLP consists of a single hidden layer with 128 units. We use 10 coupling layers to ensure sufficient expressiveness. The resulting conditional density is given by:

$$q(z \mid x) = q_\mathcal{N}(h_\nu(z; x)) \cdot |\det J|,$$

where $q_\mathcal{N}(\cdot)$ denotes the standard Gaussian density, $h_\nu(z; x)$ is the transformed latent variable via the flow, and $\det J$ is the product of the Jacobian determinants from each flow transformation.

We train LBF-NPE, MDN, and the normalizing flow using the AdamW optimizer (Loshchilov & Hutter, 2019) with a learning rate of $10^{-5}$ for 50,000 steps. The batch size is set to 1024, and training completes in approximately 2 hours. Maximum GPU memory usage is around 8400MB. For evaluation, we use a held-out set of 1000 $(z, x)$ pairs to compute the average forward and reverse KL divergences. For each test observation $x$, LBF-NPE, MDN, and the normalizing flow estimate the density $q(z \mid x)$ over a $100 \times 100$ uniform grid on $[-1, 1]^2$. These estimated posteriors are normalized such that their integral over the grid equals 1. The true posterior $p(z \mid x)$ is computed analytically over the same grid, enabling pointwise comparison. We then calculate the forward and reverse KL divergences between the estimated and true posteriors and report the average over all 1000 test cases in Table 1. For the illustrative posterior plots shown in Figure 2, we fix $x = 0.7$ and visualize the estimated density $q(z \mid x)$ from each method over the same $100 \times 100$ grid.

### D.3 2D Case Studies: Ring

The statistical model for the `ring` case study in Section 6.2 is defined as:

$$z_1, z_2 \sim \text{Unif}[-1, 1],$$
$$z = (z_1, z_2),$$
$$x \mid z \sim \mathcal{N}(\|z\|^2, \sigma^2),$$

where $\sigma^2 = 10^{-2}$. The resulting posterior, $P(z \mid x) \propto \exp\left(-(\|z\|^2 - x)^2/(2\sigma^2)\right) \cdot \mathbb{I}(-1 \le z_1, z_2 \le 1)$, forms a ring-shaped distribution in the latent space, with radius approximately $\sqrt{x}$.

The network architectures and training configurations used in this case are identical to those described in Appendix D.2. An example posterior $q(z \mid x = 0.7)$ is visualized in Figure 2.

### D.4 2D Case Studies: Spiral

The `spiral` model is defined as follows:

$$b \sim \text{Unif}[0.1, 0.5]$$
$$d \sim \text{Unif}[0.0, s_b(2\pi)]$$
$$\theta = s_b^{-1}(d)$$
$$r = b\theta$$
$$z_1 = r\cos(\theta)$$
$$z_2 = r\sin(\theta)$$
$$z = (z_1, z_2)$$
$$x \mid z \sim \mathcal{N}(b, \sigma^2)$$

where $\sigma^2 = 10^{-4}$, and $s_b(\theta) = \frac{b}{2}(\theta\sqrt{1 + \theta^2} + \sinh^{-1}(\theta))$. The posterior is $P(z \mid x) \propto \exp\left(-(\frac{r}{\theta} - x)^2/(2\sigma^2)\right) \cdot \mathbb{I}(0 \le \theta \le 2\pi, 0.1\theta \le r \le 0.5\theta)$.

Most training settings follow those in Appendix D.2, except that we increase the number of coupling layers in normalizing flow to 16. We observe minimal performance gain beyond this depth. For visualization in Figure 2, we display the estimated posterior $q(z \mid x = 0.35)$ over the $100 \times 100$ grid.

### D.5 Object Detection

We define the image generative model as follows:

$$l_1, l_2 \sim \text{Unif}([0, 16] \times [0, 16]),$$
$$f_1, f_2 \sim \mathcal{N}(\mu, \sigma^2),$$
$$I = \text{Image}(\{l_1, l_2\}, \{f_1, f_2\}, \text{PSF}),$$
$$x_{j,k} \sim \text{Poisson}(I_{j,k}),$$

where $\mu = 2000$, $\sigma^2 = 400^2$, and $\mathrm{Image}(\cdot)$ and $\mathrm{PSF}(\cdot)$ are defined below. Note that in our implementation, flux values are constrained to be positive.

---

**Algorithm 2:** `Image`

**Inputs:** list of source locations $\{l_1, l_2\}$; list
       of source fluxes $\{f_1, f_2\}$;
       point-spread function PSF.
Initialize pixel location matrix $pl$
**for** $l_i, f_i$ in zip($\{l_1, l_2\}, \{f_1, f_2\}$) **do**
    Compute relative location $rl_i = pl - l_i$
    Compute PSF density $d_i = \mathrm{PSF}(rl_i)$
    Compute $I_i = f_i \times d_i$
**end**
Compute $I = I_1 + I_2$
Return $I$.

---

**Algorithm 3:** `PSF`

**Inputs:** relative position matrix $rl_i$.
Compute
    $d_i = -(rl_i[\ldots, 0]^2 + rl_i[\ldots, 1]^2)/(2\sigma_{\mathrm{PSF}}^2)$

Compute $d_i = \exp(d_i)/\mathrm{sum}(\exp(d_i))$
Return $d_i$.

---

The pixel location matrix $pl$ is a mesh grid of shape $(H, W, 2)$ defined over $[0.5, 1.5, \ldots, H - 0.5] \times [0.5, 1.5, \ldots, W - 0.5]$, where $H$ and $W$ are the height and width of the image, respectively. Each source location $l_i$ is a 2D vector, and the term $rl_i[\ldots, 0]^2 + rl_i[\ldots, 1]^2$ is a matrix of shape $(H, W)$. We use $\sigma_{\mathrm{PSF}}^2 = 1$. Each source flux $f_i$ is a scalar. Before passing the image to the network, we normalize it using min-max scaling: $x = (x - \min(x))/(\max(x) - \min(x))$.

Since the input is a $16 \times 16$ image, we employ a convolutional layer in our network to reduce computational cost. The first layer of the model, $f_\phi$, is a 2D convolutional layer with a kernel size of 4, increasing the channel dimension from 1 to 3. This is followed by a 2D max pooling layer and a ReLU activation. The output is then flattened and passed through four MLP layers, each with 128 hidden units, layer normalization, and ReLU activations. Another model, $s_\psi$, is an MLP with four hidden layers, each with 128 units, layer normalization, and ReLU activations. The outputs of $f_\phi$ and $s_\psi$ are reparameterized via stereographic projection.

As each image contains two astronomical sources, we compute the loss separately for each source. For the source located at $l_i$, the first term in the loss is $-w\hat{f}_\phi(x)^\top \hat{s}_\psi(l_i)$. Only $\hat{s}_\psi(l_i)$ needs to be evaluated per source; shared terms such as $\hat{f}_\phi(x)$ and the integral term can be reused across both. We approximate the integral using Monte Carlo integration with 22,500 random samples. The final loss is the sum of the losses for both sources. For alternating optimization, we train one of $f_\phi$ or $s_\psi$ for 300 steps at a time (shorter than the 1000-step updates used in the 2D case studies; see Appendix D.2) since convergence is typically achieved more rapidly in this setting. Optimization is performed using the AdamW optimizer (Loshchilov & Hutter, 2019) with a learning rate of 0.001. The total number of training steps is 45,000, with overall training time under two hours.

For posterior visualization, we adopt the same procedure as in the previous 2D case studies (see Appendix D.2) but evaluate the posterior over a $200 \times 200$ grid on the domain $[0, 16]^2$. The estimated posterior for a certain image is shown in Figure 3. To generate this posterior, we leverage a model trained with $K = 64$. For results with other values of $K$ (e.g., 9, 20, 36), we provide further discussion in Appendix E.3.

### D.6 REDSHIFT ESTIMATION

Our redshift experiment extends the methodology of the Bayesian Light Source Separator (BLISS) (Liu et al., 2023; Hansen et al., 2022; Patel et al., 2025). For a given generative model of astronomical images and latent quantities (locations; fluxes; type of object; redshift; etc.), BLISS utilizes neural posterior estimation (Papamakarios & Murray, 2016) to perform amortized variational inference. The network architecture for BLISS operates on *tiles* of images, returning distributional parameters for each object detected per tile. The architecture is thus convolutional, except for several additional image normalizations and other design choices suitable for astronomical image processing.

For samples of the generative model, we use images from two tracts of the LSST DESC DC2 Simulated Sky Survey (LSST Dark Energy Science Collaboration et al., 2021; 2022), numbers 3828

and 3829. LBF-NPE does not sample the generative model on-the-fly in this setting, but only has access to a finite number of draws from the training sets.

We use the BLISS preprocessing routines to produce training, validation, and test image sets, along with ground-truth catalogs. Images, each of size $80 \times 80$, are processed in batches of 64 by the BLISS inference network, which further splits these into $4 \times 4$-pixel tiles. The network is fit to the training set to minimize the forward KL divergence using a learning rate of 0.001. All nuisance latent variables are marginalized over, and we only score redshift variational posteriors; BLISS also allows easy addition of posteriors on other latent quantities to the computed NLL loss, should the user desire to perform inference on them.

We adapt the neural network architecture from BLISS for redshift estimation. The complete architectural details and parameter configurations are provided in Table 3. As shown in the table, the input and output shapes of each layer are expressed as tuples, e.g., `(bands, h, w)` or `(64, h, w)`, where `bands` denotes the number of bands in the input astronomical images. In the DC2 dataset, there are six bands: *u, g, r, i, z, y*. The variables `h` and `w` represent the image's height and width, respectively, and are both set to 80 in our experiments. The model is composed primarily of three types of layers: `Conv2DBlock`, `C3Block`, and `Upsample`. A `Conv2DBlock` is a composite module consisting of a 2D convolution, group normalization, and a SiLU activation function. The `C3Block` is adapted from the YOLOv5 architecture (Jocher et al., 2020). It comprises three convolutional layers with kernel size 1 and includes skip connections implemented via multiple bottleneck blocks (parameterized by n). The `Upsample` layer performs spatial upscaling of the input tensor by a specified factor. The architecture follows a U-shaped design with four downsampling steps followed by two upsampling steps. To denote skip connections and input dependencies between layers, we use the "Input From" column. For example, the entry "[-1, 9]" indicates that the current layer takes as input the concatenation of the outputs from the previous layer and layer 9. The final layer is a convolutional module with kernel size 1, producing an output of shape `(n_coeff, h/4, w/4)`, where `n_coeff` is the number of predicted coefficients per tile.

The forward KL divergence framework prescribes that predictions are only scored for true objects. Accordingly, for each ground-truth redshift in the training set, we score the predicted NLL computed from the variational distribution for the 4x4 pixel tile containing that object. BLISS makes this transdimensional inference problem (a result of the number of objects per tile being unknown *a priori*) tractable by sharing parameters across objects within the same $4 \times 4$-pixel tile, at the cost of bias introduced by this approximation. For both the MDN and B-spline parameterizations, we fit to the training data for 30 epochs and use the model weights with the lowest held-out NLL on the validation set to compute the test-set NLL. Training the inference network $f_\phi$ takes approximately 12 hours on a single NVIDIA GeForce RTX 2080 Ti GPU. We note that, due to the approximations involved in using a finite training set rather than true "simulated" draws, we can easily overfit to the training and validation sets. The procedure outlined above aims to mitigate these issues to the extent possible.

| Layer # | Input From | Input Shape | Layer Type | Config | Output Shape |
|---------|------------|-------------|------------|--------|--------------|
| 1 | -1 | `(bands, h, w)` | Conv2DBlock | `in_ch=bands; out_ch=64; kernel_size=5` | `(64, h, w)` |
| 2 | -1 | `(64, h, w)` | Conv2DBlock | `in_ch=64; out_ch=64; kernel_size=5` | `(64, h, w)` |
| 3 | -1 | `(64, h, w)` | Sequence of Conv2DBlock | `in_ch=64; out_ch=64; kernel_size=5; sequence_len=3` | `(64, h, w)` |
| 4 | -1 | `(64, h, w)` | Conv2DBlock | `in_ch=64; out_ch=64; kernel_size=3; stride=2` | `(64, h/2, w/2)` |
| 5 | -1 | `(64, h/2, w/2)` | C3Block | `in_ch=64; out_ch=64; n=3` | `(64, h/2, w/2)` |
| 6 | -1 | `(64, h/2, w/2)` | Conv2DBlock | `in_ch=64; out_ch=128; kernel_size=3; stride=2` | `(128, h/4, w/4)` |
| 7 | -1 | `(128, h/4, w/4)` | C3Block | `in_ch=128; out_ch=128; n=3` | `(128, h/4, w/4)` |
| 8 | -1 | `(128, h/4, w/4)` | Conv2DBlock | `in_ch=128; out_ch=256; kernel_size=3; stride=2` | `(256, h/8, w/8)` |
| 9 | -1 | `(256, h/8, w/8)` | C3Block | `in_ch=256; out_ch=256; n=3` | `(256, h/8, w/8)` |
| 10 | -1 | `(256, h/8, w/8)` | Conv2DBlock | `in_ch=256; out_ch=512; kernel_size=3; stride=2` | `(512, h/16, w/16)` |
| 11 | -1 | `(512, h/16, w/16)` | C3Block | `in_ch=512; out_ch=256; n=3` | `(256, h/16, w/16)` |
| 12 | -1 | `(256, h/16, w/16)` | Upsample | `scale=2; mode="nearest"` | `(256, h/8, w/8)` |
| 13 | [-1, 9] | `(512, h/8, w/8)` | C3Block | `in_ch=512; out_ch=256; n=3` | `(256, h/8, w/8)` |
| 14 | -1 | `(256, h/8, w/8)` | Upsample | `scale=2; mode="nearest"` | `(256, h/4, w/4)` |
| 15 | [-1, 6] | `(384, h/4, w/4)` | C3Block | `in_ch=384; out_ch=256; n=3` | `(256, h/4, w/4)` |
| 16 | -1 | `(256, h/4, w/4)` | Conv2D | `in_ch=256; out_ch=n_coeff; kernel_size=1` | `(n_coeff, h/4, w/4)` |

Table 3: Neural network architecture for redshift estimation.

### D.7 Angular Distance Optimization

As discussed in Section 4.4, our method can be interpreted as performing angular-distance optimization, but with a loss and gradient derived from a probabilistic space. This interpretation becomes evident if we decouple the magnitude and directional components of the output tensors $f_\phi(x), s_\psi(z) \in \mathbb{R}^K$ through normalization techniques such as L2 normalization or stereographic projection reparameterization. Angular distance optimization is a common objective in modern machine learning pipelines, improving performance and ensuring consistent alignment between training and testing metrics. Several widely-used loss functions, including the triplet loss (Hoffer & Ailon, 2015), N-pair loss (Sohn, 2016), and multi-similarity loss (Wang et al., 2019), incorporate angular distance in their formulation. Cosine-based softmax loss functions are widely used for face recognition (Liu et al., 2017; Wang et al., 2018; Deng et al., 2019), and many contrastive learning algorithms (Chen et al., 2020; Tian et al., 2020; Ye et al., 2019) employ angular objectives to maximize the cosine similarity between embeddings from positive pairs.

Our variational objective in Equation (6) is related to cosine-based softmax loss, whose general form is

$$
L = -wS_{i,y_i} + \log\left(\exp(wS_{i,y_i}) + \sum_{j \neq y_i} \exp(wS_{i,j})\right), \tag{18}
$$

as described in Section 4.4, and suggesting a more general form for our variational objective, namely

$$
\hat{L}_{\text{LBF-NPE}}(\phi, \psi) = \mathbb{E}_{p(z,x)}\left[-w\hat{f}_\phi(x)^\top \hat{s}_\psi(z) + \log\left(\int \exp\left(w\hat{f}_\phi(x)^\top \hat{s}_\psi(z')\right) dz'\right)\right], \tag{19}
$$

where again $\hat{f}_\phi(\cdot)$ and $\hat{s}_\psi(\cdot)$ are normalized outputs of neural networks (i.e., with unit norm) and $w$ is a scaling factor.

The key differences from the cosine-based softmax formulation are: (1) the summation is replaced by an integral over the continuous latent space, and (2) the angular distance is computed between coefficient vectors and basis functions, rather than between learned embeddings. This connection offers two main advantages. First, our theoretical guarantees on convexity and convergence may extend to angular distance optimization problems, suggesting broader applicability. Second, our method can leverage off-the-shelf improvements developed for angular optimization, such as SEC (Zhang et al., 2020), which regularizes gradient updates to stabilize and accelerate training. Given that even simple stereographic normalization already yields strong empirical results, we leave the integration of these enhancements to future work.

In our experiments, we use stereographic reparameterization to project the output tensor onto the unit hypersphere. Precisely, $u \in \mathbb{R}^{K-1}$ is transformed to $\mathbb{R}^K$ via

$$
y = \left(\frac{2u}{1 + \|u\|^2}, \frac{1 - \|u\|^2}{1 + \|u\|^2}\right), \tag{20}
$$

ensuring that $\|y\| = 1$. This projection serves as a smooth and bijective transformation from Euclidean space $\mathbb{R}^{K-1}$ onto the $K-1$-sphere $S^{K-1} = \{x \in \mathbb{R}^K : \|x\| = 1\}$. Although this transformation changes the form of the variational objective in the neural network outputs, and thus violates some assumptions of our NTK framing from the perspective of convexity, strong empirical results suggest the benefits of reparameterization, and also the importance of future work in understanding the success of neural posterior estimation (NPE) techniques under arbitrary parameterizations. We hypothesize that the advantageous properties of this normalization stem from the smooth gradient trajectories and mapping to a compact space, discussed in more detail below.

We illustrate the stereographic normalization process in a 2D case, as shown in Figure 11. In this setting, a scalar input $u \in \mathbb{R}^1$ is projected onto a vector $y \in \mathbb{R}^2$ lying on the 1-sphere (i.e., the unit circle). For any given $u$, there exists a unique line connecting the point $(u, 0)$ and the north pole $N = (0, 1)$. This line intersects the 1-sphere at a single point, which serves as the projection of $u$. By drawing a vector from the origin to this intersection point, we obtain a unit vector $y$ on the 1-sphere. Notably, the location of the intersection reflects the magnitude of $u$: if $\|u\| > 1$, the intersection lies on the upper half of the circle; if $\|u\| < 1$, it falls on the lower half.

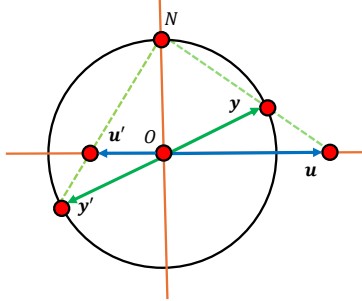

Figure 11: Visualization of stereographic projection in 2D. A scalar $u$ is mapped to a point on the unit circle via intersection of the line connecting $(u, 0)$ and the north pole $N = (0, 1)$.

This reparameterization offers several advantages. First, the stereographic projection is differentiable everywhere and provides well-behaved gradients throughout the domain. Second, the projection naturally enforces unit-norm constraints without requiring additional normalization layers or manual clipping, thus making training more stable and efficient.

# E  ADDITIONAL EXPERIMENTAL RESULTS

## E.1  EFFECT OF NORMALIZATION

We compare the neural network's output basis functions with and without stereographic projection normalization, demonstrating that normalization helps the network learn clearer boundaries between regions of the parameter space. Figure 12 and Figure 13 show the values of the 20-dimensional basis functions (i.e., $[s_\psi(z)]_i$, where $i \in \{1, 2, \ldots, 20\}$) evaluated over the plane $z \in [-3, 3] \times [-3, 3]$ for the spiral case study. It is evident that the model with normalization exhibits more distinguishable and structured partition boundaries in $z$-space, while the model without normalization suffers from blurry transitions and over-exposure artifacts, as seen in Figure 13. This highlights a key drawback of the non-normalized approach: it struggles to effectively disentangle the parameter space. Normalization also enhances interpretability by promoting better separation among basis functions. The estimated posterior density is expressed as a weighted linear combination of these basis function densities. For a given target spiral, the neural network increases the weights (i.e., $[f_\phi(x)]_i$) for dimensions whose corresponding basis functions have high overlap with the target density, and decreases weights for dimensions with low overlap.

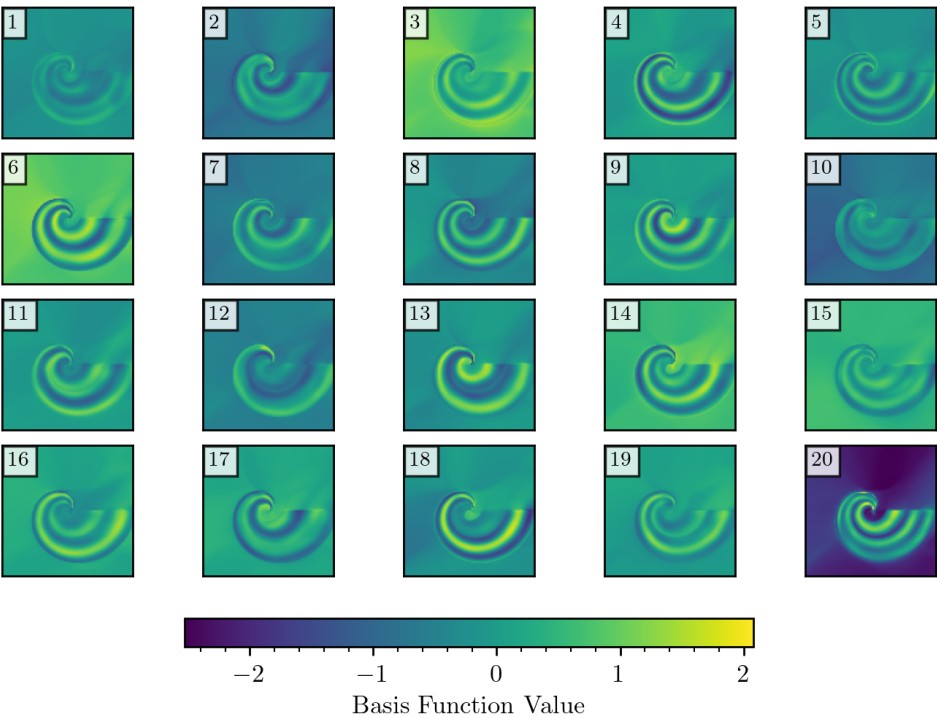

Figure 12: Density plot of 20-dim basis function (w/ stereographic projection normalization) over plane $z$. Each subplot represents the density plot of a certain dimension.

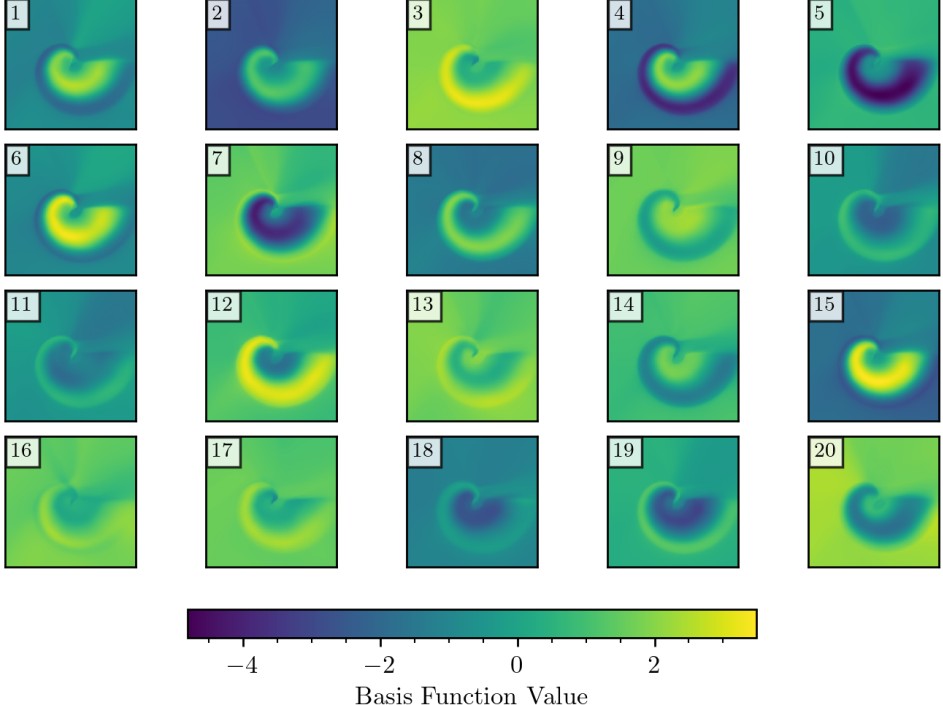

Figure 13: Density plot of 20-dim basis function (w/o stereographic projection normalization) over plane $z$. Each subplot represents the density plot of a certain dimension.

### E.2 POSTERIOR COMPARISON

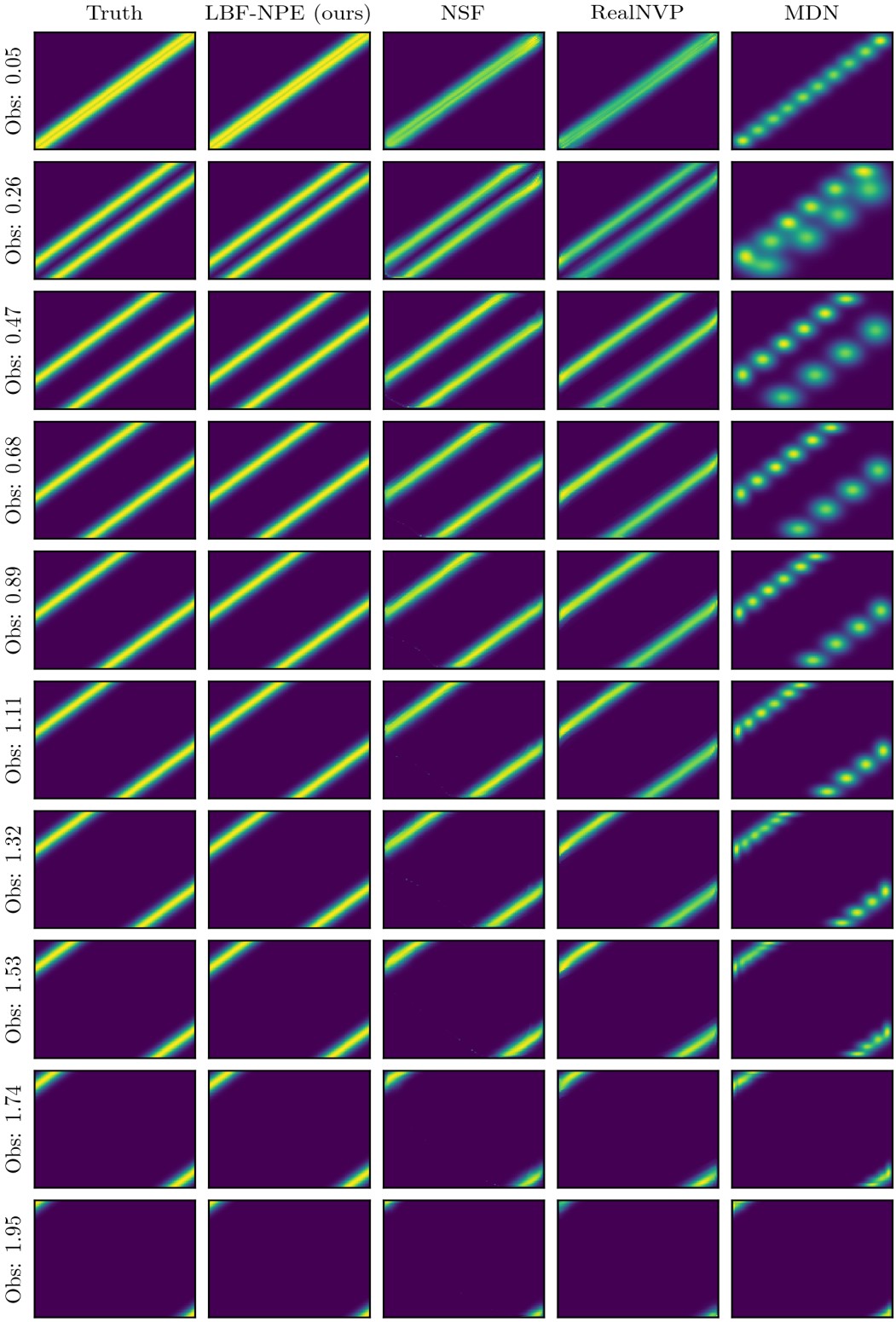

Figure 14: Visual comparison of posterior densities estimated by different methods (LBF-NPE, Normalizing Flow, MDN) against ground truth for ten representative observations.

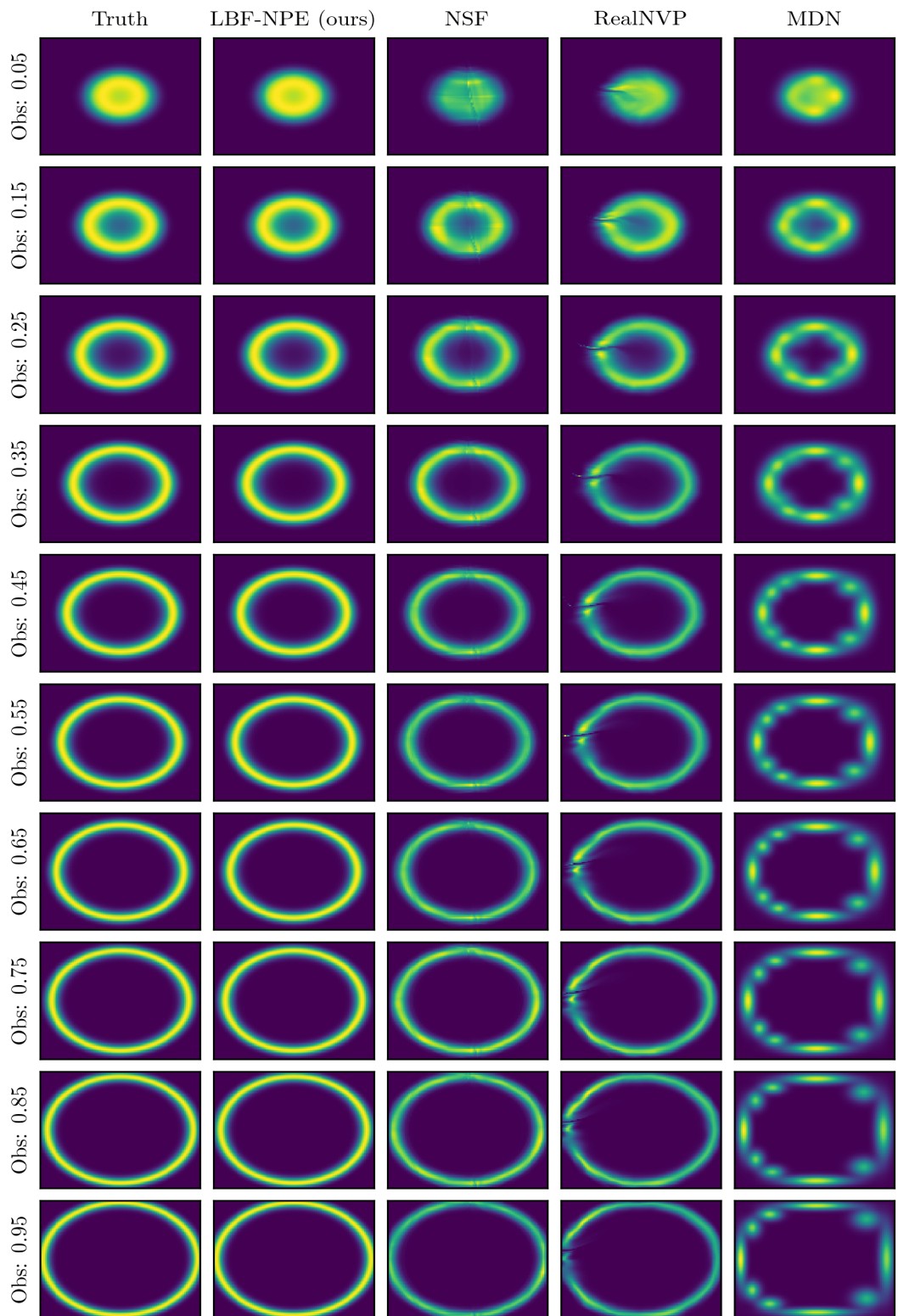

Figure 15: Visual comparison of posterior densities estimated by different methods (LBF-NPE, Normalizing Flow, MDN) against ground truth for ten representative observations.

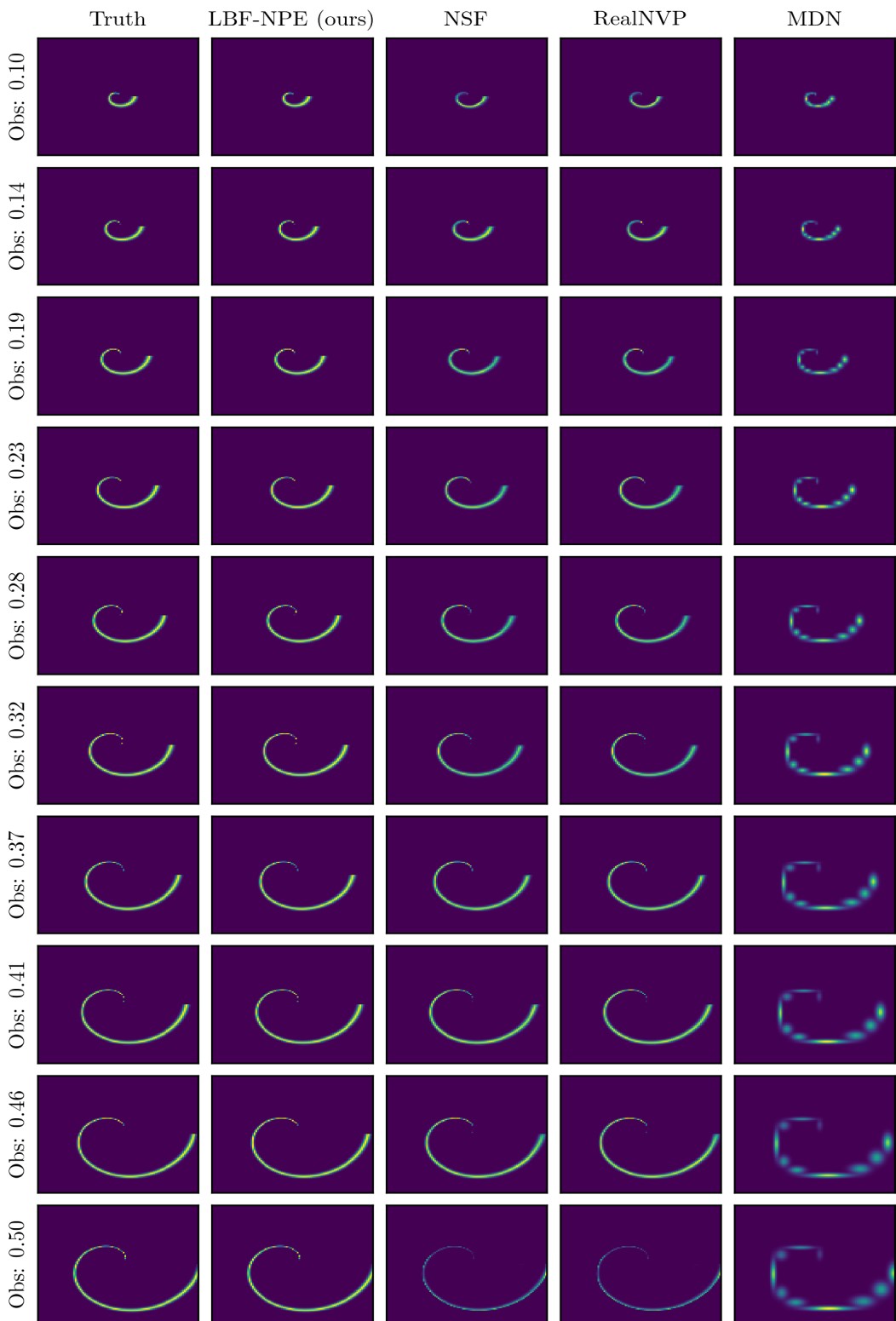

Figure 16: Visual comparison of posterior densities estimated by different methods (LBF-NPE, Normalizing Flow, MDN) against ground truth for ten representative observations.

### E.3 DIMENSIONS OF BASIS FUNCTIONS & FLEXIBILITY

The flexibility of our method is positively correlated with the dimensionality of the basis functions. We demonstrate this by analyzing both the forward and reverse KL divergences, as well as basis-function density plots, for models using basis functions of varying dimensions in the object detection case study. To quantify this, we compute the forward and reverse KL divergence between the estimated posterior distribution and a target mixture of Gaussians defined by the true underlying locations $l_1, l_2$:

$$0.5\,\mathcal{N}(l_1, 0.1^2) + 0.5\,\mathcal{N}(l_2, 0.1^2). \tag{21}$$

The results, shown in Table 4, indicate that both forward and reverse KL divergences decrease as the dimensionality of the basis functions increases. For example, the 64-dimensional basis functions achieve the lowest forward KL divergence (1.524), while the 9-dimensional basis functions result in the highest (3.397). A similar trend holds for reverse KL divergence. However, the marginal gain from increasing dimensionality diminishes as the number of basis functions grows. Increasing from 9 to 20 dimensions yields a significant improvement in forward/reverse KL divergence (1.187/0.511), but the improvement from 36 to 64 dimensions is relatively minor (0.246/0.287). This suggests that, for a complex task like object detection, a basis function dimensionality under 100 is sufficient to achieve near-optimal performance.

|  | 9-dim | 20-dim | 36-dim | 64-dim |
|---|---|---|---|---|
| **Forward KL Divergence** | 3.397 | 2.210 | 1.770 | **1.524** |
| $\triangle$ **Forward KL Divergence** | - | **-1.187** | -0.440 | -0.246 |
| **Reverse KL Divergence** | 2.380 | 1.869 | 1.360 | **1.073** |
| $\triangle$ **Reverse KL Divergence** | - | **-0.511** | -0.509 | -0.287 |

Table 4: Object detection: forward/reverse KL divergence for models of different basis function dimensions.

The basis function density plots provide further intuition for this trend. As shown in Figures 17 to 20, the 64- and 36-dimensional basis functions can partition the $z$-space into fine-grained regions, capturing detailed structure. In contrast, 20- and 9-dimensional basis functions fail to do so, resulting in coarser approximations and reduced representational capacity.

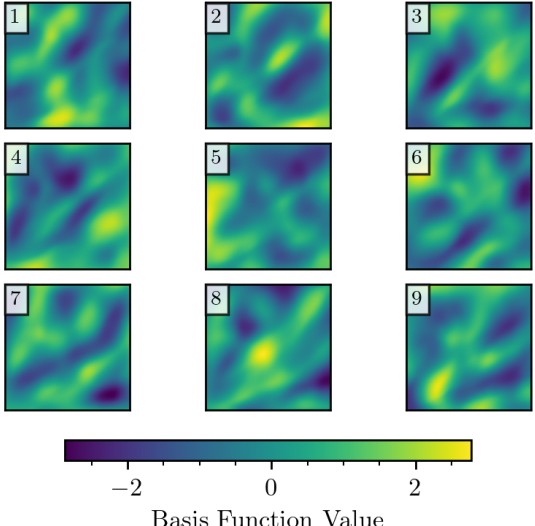

Figure 17: Object detection: density plot of 9-dim basis function over plane $z$. Each subplot represents the density plot of a certain dimension.

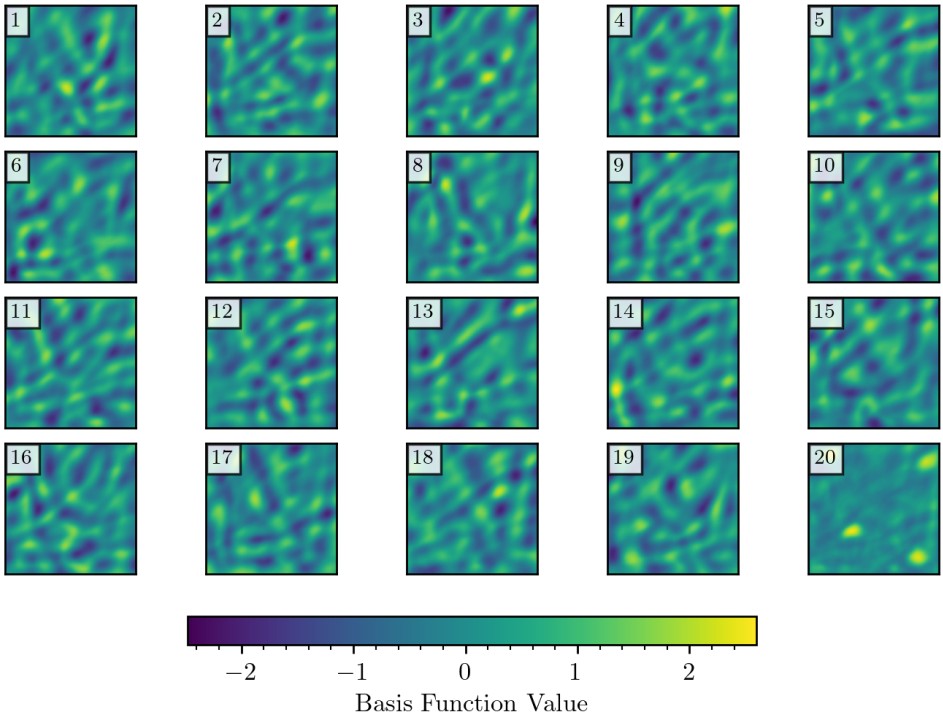

Figure 18: Object detection: density plot of 20-dim basis function over plane $z$.

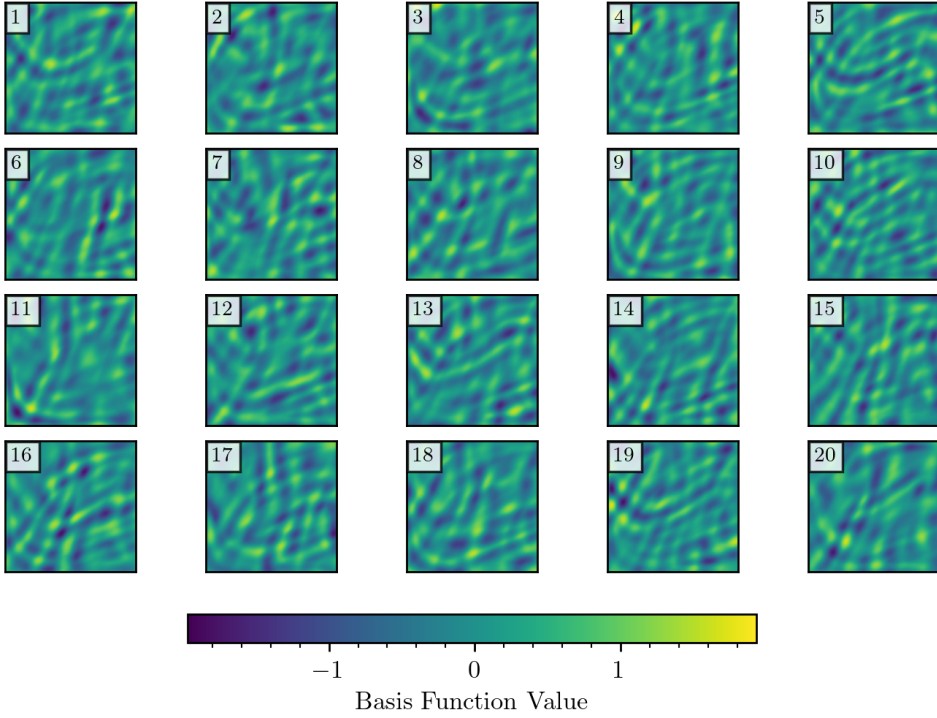

Figure 19: Object detection: density plot of 36-dim basis function over plane $z$. For brevity, we only show the first 20 dimensions.

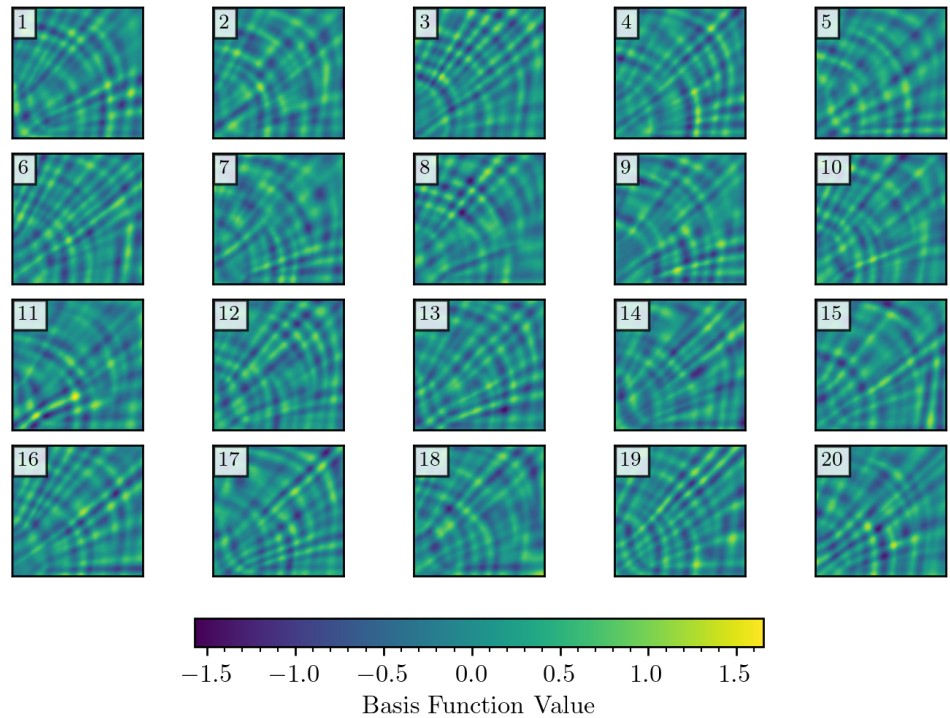

Figure 20: Object detection: density plot of 64-dim basis function over plane $z$. For brevity, we only show the first 20 dimensions.

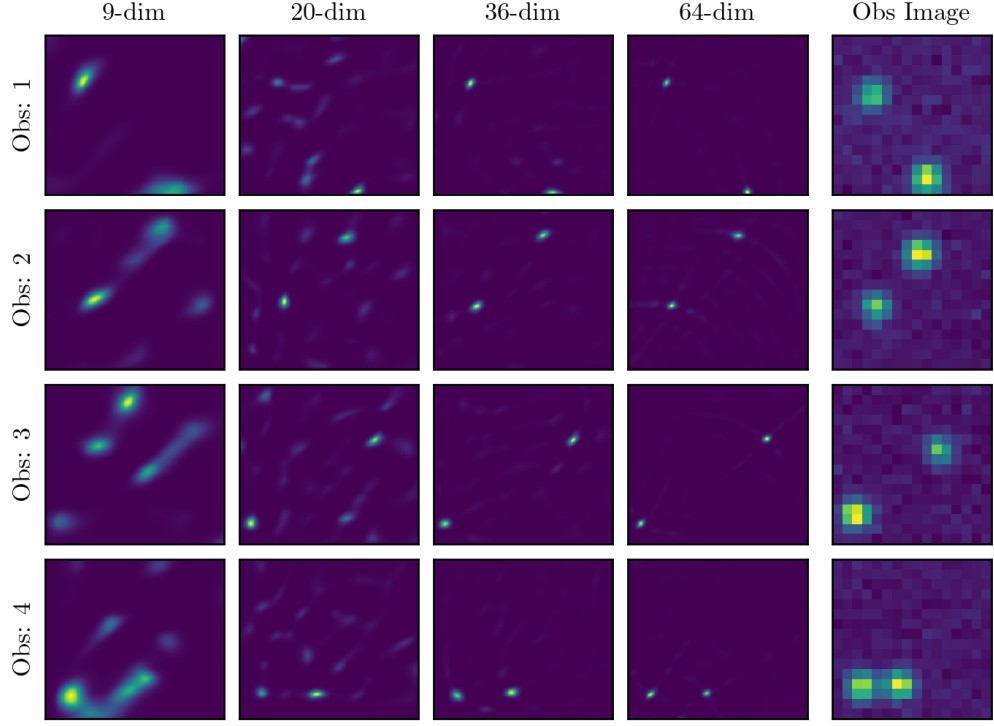

Figure 21: Object detection: 9/20/36/64-dim basis functions and the corresponding estimated posterior density.

Interestingly, in the ring case study, we observe that even a 2-dimensional basis function is sufficient to recover the ring-shaped posterior. As shown in Figure 25, the estimated posterior using a 2-dimensional basis function is visually nearly indistinguishable from the true posterior, with only minor artifacts appearing when the observation $x$ approaches extreme values (e.g., $x = 1.95$). This observation is quantitatively supported by the KL divergence results in Table 5, where both forward and reverse KL values are low (0.032/0.031) for the 2-dimensional case. These results demonstrate that for simpler posterior structures, our method can achieve accurate inference with remarkably low-dimensional basis functions.

|  | 2-dim | 4-dim | 9-dim | 20-dim |
|---|---|---|---|---|
| **Forward KL Divergence** | 0.032 | 0.0057 | 0.0056 | **0.0054** |
| $\triangle$ **Forward KL Divergence** | - | **-0.0263** | -0.0001 | -0.0002 |
| **Reverse KL Divergence** | 0.031 | 0.0032 | 0.0028 | **0.0027** |
| $\triangle$ **Reverse KL Divergence** | - | **-0.0278** | -0.0004 | -0.0001 |

Table 5: Ring: forward/reverse KL divergence for models of different basis function dimensions.

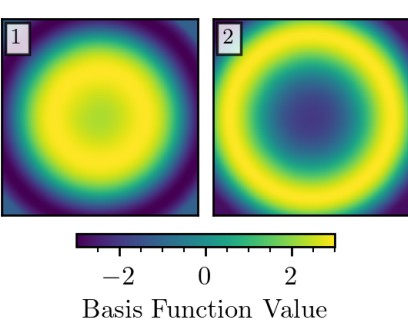

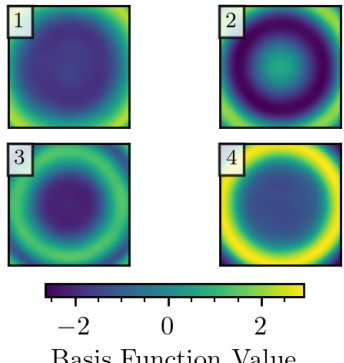

Figure 22: Ring: density plot of 2-dim basis function over plane $z$. Each subplot represents the density plot of a certain dimension.

Figure 23: Ring: density plot of 4-dim basis function over plane $z$.

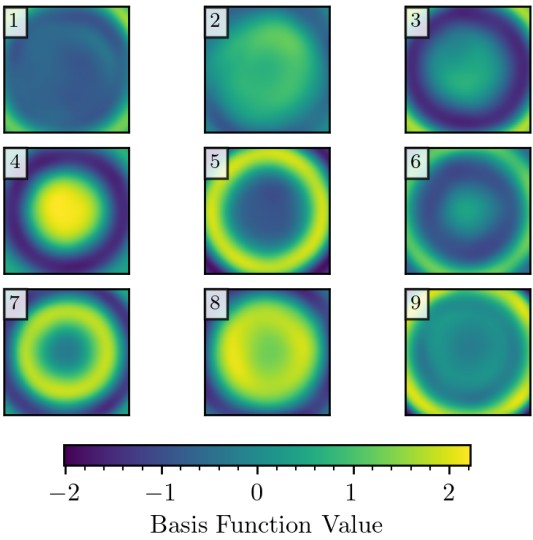

Figure 24: Ring: density plot of 9-dim basis function over plane $z$.

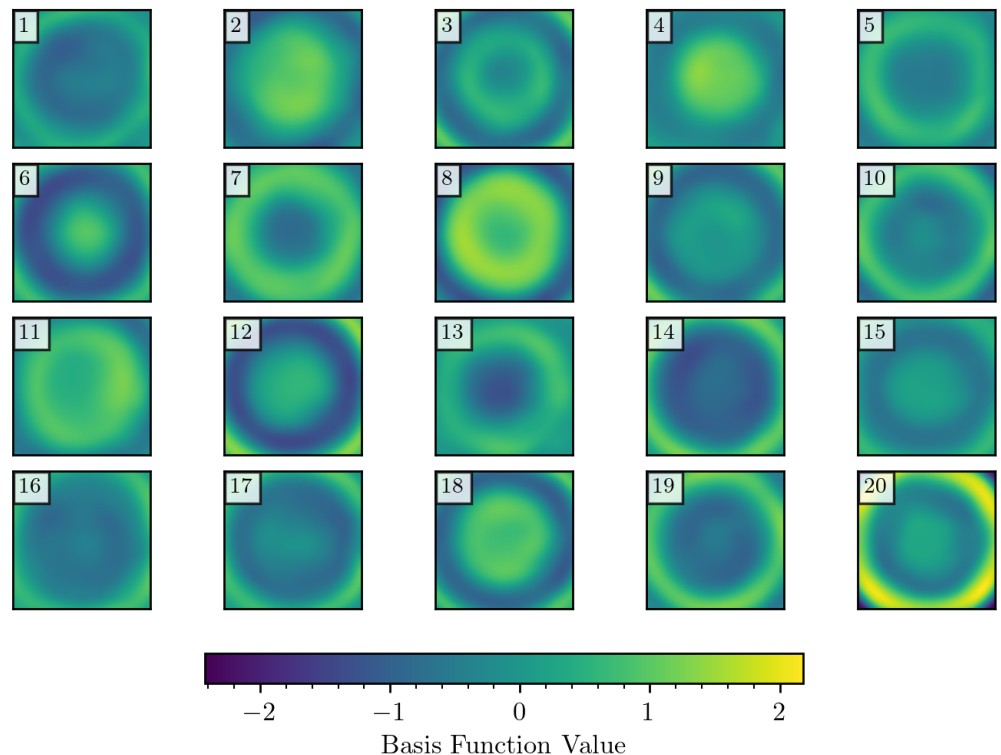

Figure 25: Ring: density plot of 20-dim basis function over plane $z$.

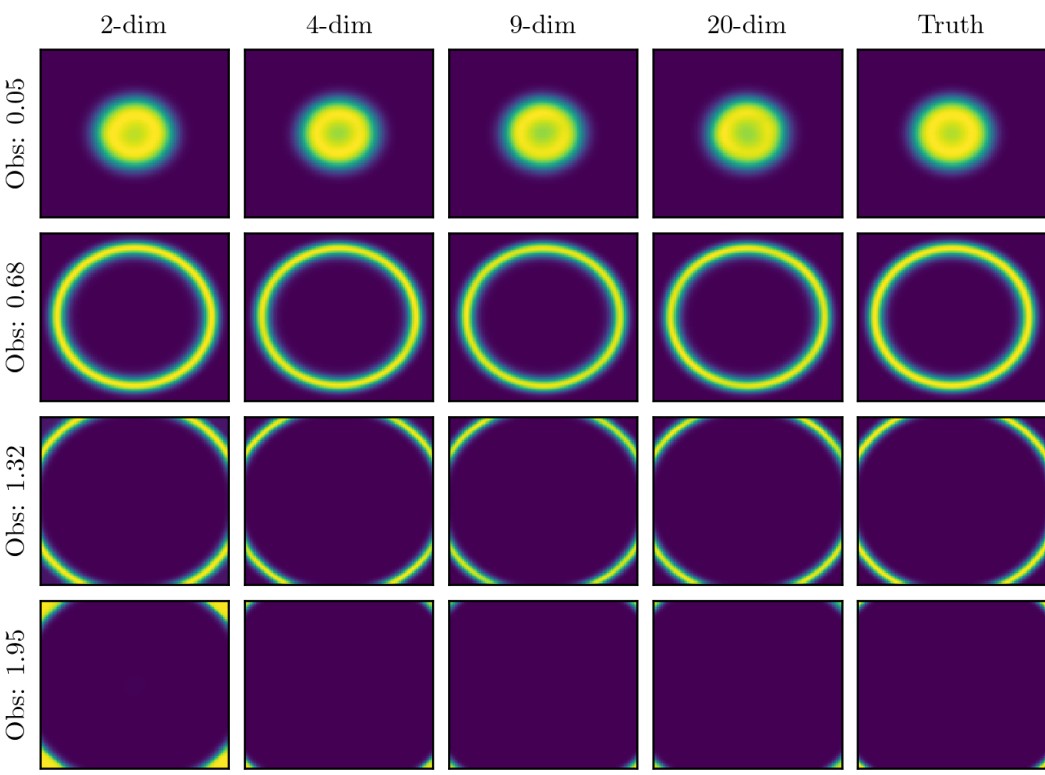

Figure 26: Ring: 2/4/9/20-dim basis functions and the corresponding estimated posterior density.

### E.4 TOWARDS HIGH DIMENSION

To evaluate the capability of our model in predicting high-dimensional posteriors, we construct a 50-dimensional annulus model defined as:

$$z_1, z_2, \ldots, z_{50} \sim \text{Unif}[0, 1],$$
$$z = (z_1, \ldots, z_{50}),$$
$$x \mid z \sim \mathcal{N}(\|z\|^2, \sigma^2),$$

where $\sigma^2 = 10^{-2}$.

In Figures 27 and 28, we randomly select two pairs of dimensions (3, 14) and (43, 47), and visualize the estimated posterior over these subspaces, i.e., $q(z_3, z_{14} \mid x)$ and $q(z_{43}, z_{47} \mid x)$. We discretize each pair onto a $100 \times 100$ grid and perform Monte Carlo integration over the remaining 48 dimensions to obtain an estimate of the posterior in the chosen 2D subspace. The results show that the estimated posteriors closely match the true posteriors, with minor discrepancies likely due to Monte Carlo integration variance. The final two columns of each plot show the marginal densities, demonstrating that our model captures the true marginal posterior distributions. This good performance can also be verified by quantitative metrics. Our model attains 0.018 forward KL divergence and 0.022 reverse KL divergence, on average, across 50 dimensions.

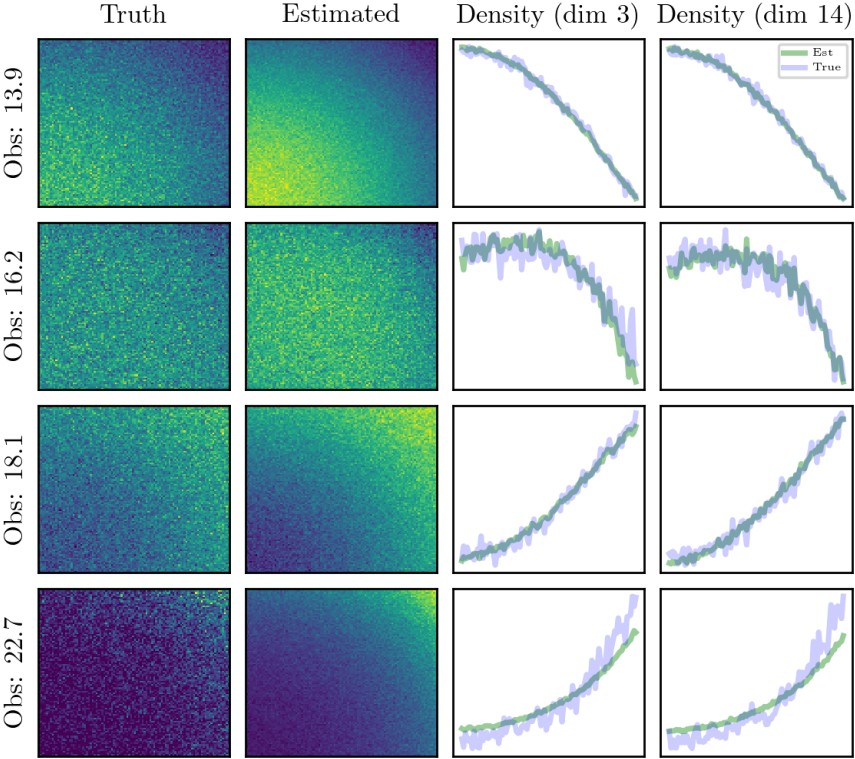

Figure 27: True and estimated posterior density over dimensions 3 and 14. The y-axis in the last two columns represents the marginal density.

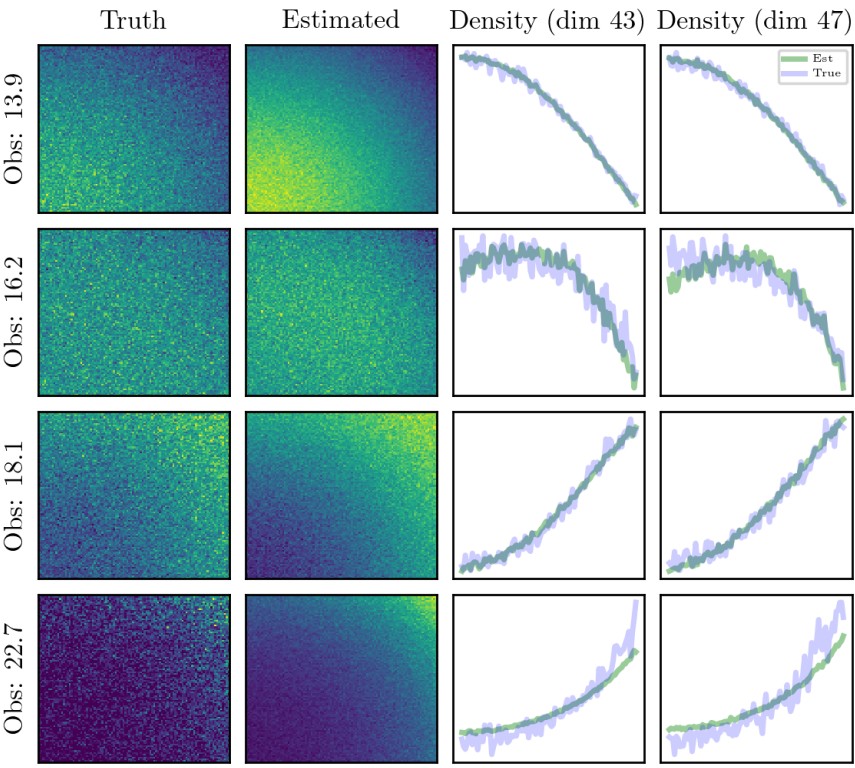

Figure 28: True and estimated posterior density over dimensions 43 and 47. The y-axis in the last two columns represents the marginal density.

### E.5 EigenVI on 2D Case Studies

We evaluate EigenVI (Cai et al., 2024) on three two-dimensional targets with thin or curved posterior density patterns. We use a tensor product expansion with $K = 16$ basis functions per axis ($K^2 = 256$ coefficients) and $50,000$ importance samples for fitting. The reconstructions capture only coarse structure: for the diagonal bands, EigenVI recovers orientation, but the two ridges are blurry; for the ring, it fills the central hole and collapses mass inward; for the spiral, it loses the manifold and yields blurry lobes. These failures arise from the spectral bias of orthogonal expansions, which under-represent the high-frequency content required by a thin or strongly curved posterior. While increasing $K$ can help, computational and statistical costs scale as $K^d$ (here $d = 2$), making adequate resolution impractical. In sum, with practical $K$, EigenVI is adequate for smooth densities but inadequate for multimodal or topologically nontrivial two-dimensional targets.

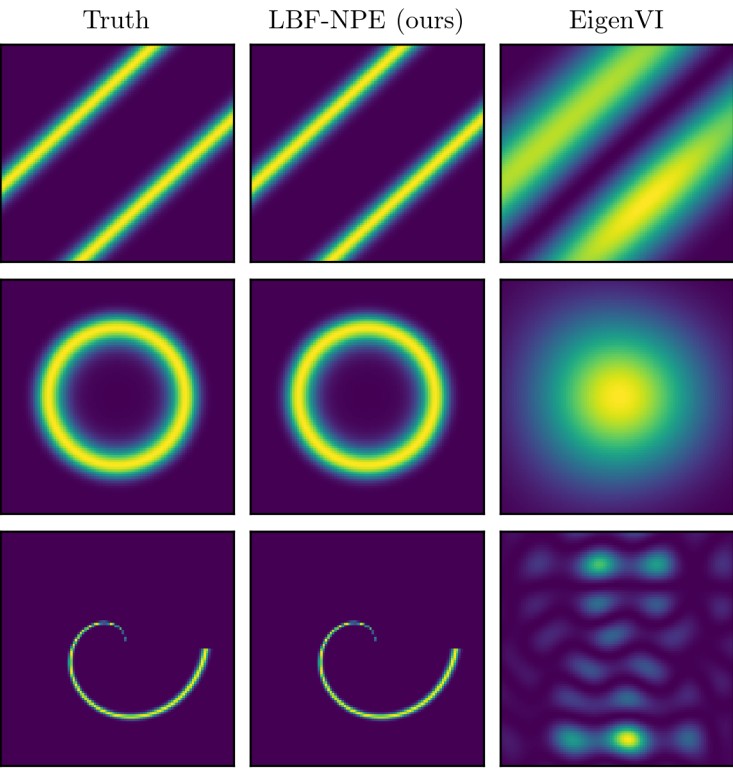

Figure 29: EigenVI results for three 2D test cases. Along each axis, we fit 16 basis functions (i.e., $K = 16$ in their original paper); For their importance sampling, we draw 50,000 samples.

### E.6   SCORE MATCHED NEURAL EXPONENTIAL FAMILIES ON 2D CASE STUDIES

We reproduce the method proposed in Score Matched Neural Exponential Families for Likelihood-Free Inference (Pacchiardi & Dutta, 2022) and evaluate its sampling quality on three two-dimensional case studies. Their approach estimates the unnormalized probability $p(x \mid z)$, rather than our focus on $p(z \mid x)$, and employs Exchange MCMC to draw posterior samples. As illustrated in the figure below, the Exchange MCMC samples are suboptimal, often overdispersed and misaligned with the true density. In the bands case study, the samples fail to align with the ridges and instead spread into low-density regions. In the ring case study, many samples are scattered inside the ring rather than concentrating on its boundary, where the density peaks. In the spiral case study, the bias is most evident, with samples deviating substantially from the high-density spiral structure. We also apply inverse transform sampling, which was not considered in the original paper. This approach produces samples that more closely follow the true density across all three case studies, though in the spiral example, a residual bias remains.

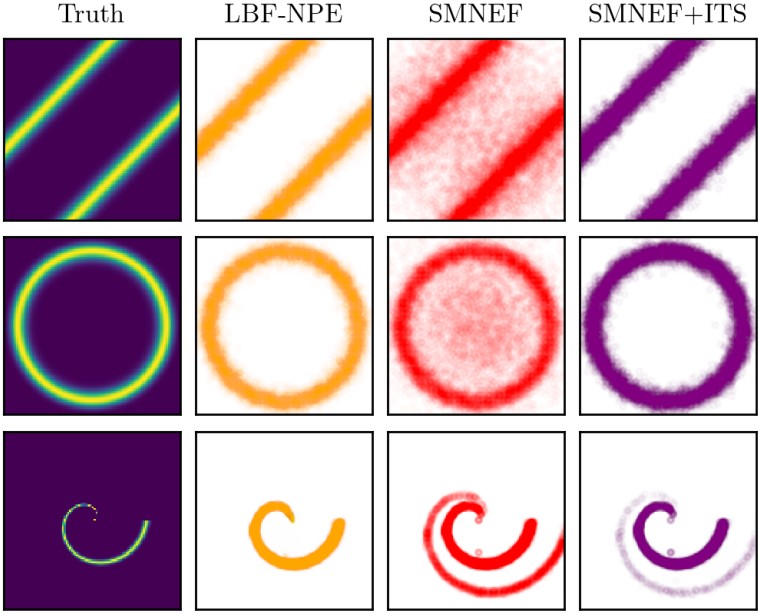

Figure 30: Sampling results of Score Matched Neural Exponential Families (SMNEF) for Likelihood-Free Inference on three 2D case studies: SMNEF uses Exchange MCMC as its default sampling setting. SMNEF+ITS is our variant of SMNEF, customized for low-dimensional settings. For our method, we use inverse transform sampling.

### E.7 COMPUTATIONAL COST

Table 6 compares the computational efficiency of several baseline models with the proposed LBF-NPE across the 2D case studies, object detection, and redshift estimation. While LBF-NPE shows slightly higher per-step runtime and memory usage than some baselines, its principal advantage is its substantially faster convergence, requiring markedly fewer training steps, such as 8k in the 2D case and 48k in the redshift task. This accelerated convergence results in competitive or superior total training time across tasks, indicating that LBF-NPE achieves an effective optimization trajectory. A notable detail is that, in the redshift experiments, the computational costs of all three methods appear very similar. This is largely due to the dominant overhead from the convolutional U-shape network used for image processing, which outweighs differences in the loss computation. Another point of clarification is that the GPU memory usage reported here is lower than the values in Section D.2 (e.g., the peak usage is approximately 8400 MB). This discrepancy arises because, for the computational cost evaluation, we disable GPU memory preallocation in JAX, which otherwise reserves roughly 75% of the available GPU memory.

| Case study | Method | Time per step (s/step) | GPU Memory (MB) | Converge at (steps) | Time until converge (s) |
|---|---|---|---|---|---|
| **Sinusoidal likelihood (batch size: 1024)** | LBF-NPE | 0.038 | 312 | **4k** | **152** |
| | MDN | **0.021** | **248** | 10k | 210 |
| **2D case studies (batch size: 1024)** | LBF-NPE | 0.127 | 970 | **8k** | 1016 |
| | NSF | 0.084 | 780 | 29k | 2436 |
| | RealNVP | 0.082 | 690 | 25k | 2050 |
| | MDN | **0.044** | **430** | 13k | **572** |
| **Object detection (batch size: 1024)** | LBF-NPE | 0.143 | 2230 | 15k | 2145 |
| **Redshift (batch size: 32)** | LBF-NPE | 0.28 | 7319 | **48k** | **13440** |
| | NSF | 0.28 | 7012 | 80k | 22400 |
| | MDN | **0.26** | **6988** | 54k | 14040 |

Table 6: Computational cost: For 2D case studies, we only report the computational cost for the `spiral` case study, because the other two case studies have similar computational costs. The "Converge at (steps)" refers to the maximum number of training steps required to achieve the performance reported in the paper. NSF is the abbreviation of Neural Spline Flow.

### E.8 BAYESIAN NEURAL NETWORKS AS GENERATIVE MODELS

A Bayesian neural network (BNN) is a neural network in which each weight (and bias) is treated as a probability distribution rather than a fixed value (MacKay, 1992a;b; Neal, 1996). When making predictions, it marginalizes over these distributions to produce not just a prediction, but also an estimate of uncertainty. We consider BNNs as generative models to illustrate that LBF-NPE can perform posterior predictive inference while implicitly marginalizing over a high-dimensional parameter space. In particular, we consider a two-layer fully connected BNN:

$$\text{BNN}_\theta(x) = \text{Linear}_{\theta_2}\big(\text{ReLU}(\text{Linear}_{\theta_1}(x))\big), \qquad \theta_1, \theta_2 \sim \mathcal{N}(\mathbf{0}, \mathbf{I}), \tag{22}$$

where $\theta_1$ and $\theta_2$ are the weight matrices of the first and second linear layers, respectively, and $\text{ReLU}(\cdot)$ denotes the rectified linear unit activation. The network takes a one-dimensional input and produces a one-dimensional output through a hidden layer of width 16. Throughout this experiment we restrict attention to $x \in [0, 10]$ and $y \in [-8, 8]$.

Let $\theta = (\theta_1, \theta_2)$ and let the dataset be

$$\mathcal{D} = \{(x_i, y_i)\}_{i=1}^n, \qquad y_i = \text{BNN}_\theta(x_i) + \epsilon_i, \quad n = 5,$$

for a fixed draw of $\theta$ from $\mathcal{N}(\mathbf{0}, \mathbf{I})$, and noise $\epsilon_i \sim \mathcal{N}(0, 1)$. The posterior predictive distribution for a new pair $(x', y')$ is

$$p(y' \mid x', \mathcal{D}) = \int p(y' \mid x', \theta)\, p(\theta \mid \mathcal{D})\, \mathrm{d}\theta, \tag{23}$$

which involves integration over the high-dimensional weight vector $\theta$. Our goal in this section is to show that LBF-NPE can approximate the conditional density in (23) without ever explicitly sampling or optimizing over $\theta$.

In our implementation, LBF-NPE parameterizes the conditional density via a basis-function network $s_\psi$ and a coefficient network $f_\phi$. The coefficient network $f_\phi$ takes as input the query variate $x'$ together with the conditioning set $\mathcal{D}$, while the basis-function network $s_\psi$ takes $y'$ as input. Together they define

$$\text{LBF-NPE}(y', x', \mathcal{D}) \approx p(y' \mid x', \mathcal{D}),$$

and are trained using Algorithm 1. Both $s_\psi$ and $f_\phi$ are implemented as four-layer multilayer perceptrons with 128 hidden units per layer; the resulting basis-function and coefficient vectors have dimension 20. We optimize the parameters $(\psi, \phi)$ with the AdamW optimizer (Loshchilov & Hutter, 2019), using a learning rate of $10^{-3}$ for 10,000 gradient steps.

The qualitative behavior of the learned posterior predictive distributions is shown in Figure 31. Across the four panels, the underlying BNN functions (orange curves) exhibit markedly different slopes, curvatures, and ranges, yet LBF-NPE recovers their overall shape from only $n = 5$ observations. Moreover, the posterior highest-density interval (HDI) is wide in regions with sparse observations and narrow in regions with many data points. For example, when no observation is available near $x \in [0, 2]$ (top-left panel), the HDI is wide, whereas in regions densely populated with observations (e.g., $x \in [2.5, 5.0]$ in the bottom-left panel) the HDI becomes narrow. This pattern indicates that LBF-NPE successfully captures the epistemic uncertainty induced by marginalization over the BNN weights.

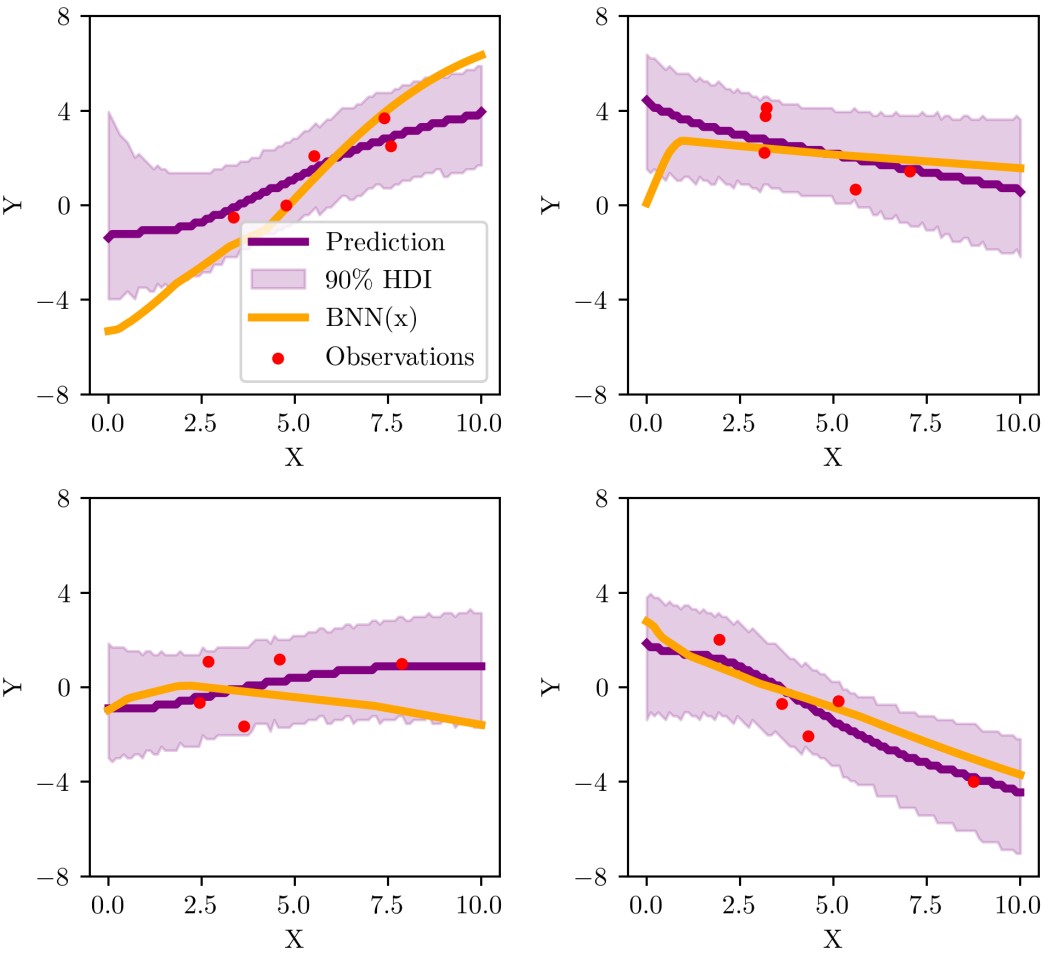

Figure 31: Posterior predictive distributions obtained from LBF-NPE on four synthetic regression problems generated by a Bayesian neural network (BNN). Each panel corresponds to a different draw of the BNN weights. The orange curve is the true mapping $x \mapsto \mathrm{BNN}_\theta(x)$, and the red dots denote the $n = 5$ observed data points used for inference. The solid purple curve shows the pointwise mode of the posterior predictive density produced by LBF-NPE, and the shaded region indicates the associated $90\%$ highest–density interval (HDI).

