# OpenReview forum: "Neural Posterior Estimation with Latent Basis Expansions"
_ICLR.cc/2026/Conference — ICLR 2026 Poster_

### Official Review · Reviewer_hPZ3 · 2025-10-30

**Soundness:** 3
**Presentation:** 3
**Contribution:** 3
**Rating:** 6
**Confidence:** 2

**Summary:**

The paper proposes a new variational family for Neural Posterior Estimation (NPE) that parameterizes the log-density of the variational distribution through basis expansions of latent variables. Instead of the traditional mixture of Gaussian or flow-based families, the authors parametrize the log density as a linear combination of latent basis functions, with coefficients modeled by an inference network and basis function network either fixed (e.g., B-splines, wavelets) or learned adaptively. The authors have claimed convexity and stable optimization in the fixed-basis case, and flexibility and expressivity comparable to normalizing flows in the adaptively learning case. The proposed method shows strong empirical results across both synthetic and real-world applications, consistently outperforming baselines such as MDN, RealNVP, and NSF on KL divergence and NLL.

**Strengths:**

1. The latent basis function formulation for the variational family is original, offering a middle ground between the simple Gaussian family and black-box flows.
2. Convexity and convergence properties are explicitly proven (App. B) and related to NTK theory, a rare level of rigor for NPE papers. The fixed-basis variant provides stable inference under certain conditions where global convexity is crucial.
3. Benchmarks cover synthetic, multimodal, and real astrophysical datasets, including redshift estimation (DC2 dataset). The proposed LBF-NPE achieves lower forward/reverse KL and NLL than baselines.

**Weaknesses:**

1. Most experiments are in ≤ 2D latent spaces. The high-dimensional example (App. E.4) is also synthetic (50-dimensional); stronger evidence on real higher-dimensional tasks would strengthen the claim of scalability.
2. While the impact of basis dimensionality and normalization is explored (App. E.1–E.3), ablations on the effect of stereographic projection and alternating optimization are somewhat fragmented.
3. Given that both the basis function $s_\psi$ and inference network $f_\phi$ are parameterized as deep neural networks, computational efficiency remains a concern for this algorithm.
4. Although Algorithm 1 provides a gradient estimator for the objective function, it lacks pseudocode for the entire algorithm. Including this would clarify the overall process.

**Questions:**

1. Could the authors elaborate on whether LBF-NPE can be extended to high-dimensional latent spaces (e.g., posteriors of Bayesian neural networks) without prohibitive integration cost? Given time constraints, you can illustrate this only using a small toy regression experiment.
2. How does the computational efficiency of the proposed LBF-NPE compare to other baselines? Can you provide inference time comparisons?
3. The current gradient estimator is biased but consistent. Have you quantified this bias, or explored control variates or reparameterization tricks to reduce it?
4. While Proposition 1 ensures convexity when either $s_\psi$ or $f_\phi$ is fixed, what guarantees or empirical evidence exist when both networks are trained jointly? Have the authors observed non-convex behavior or mode collapse in practice?
5. Can you provide quantitative results comparing fixed (B-spline/wavelet) versus adaptive bases in terms of convergence speed, stability, and overfitting? How do you decide whether to use B-splines, wavelets, or neural bases across different tasks? Is there an automatic basis selection mechanism?
6. The choice of the scaling factor $w$ is ad-hoc. Is it tuned per task, or could it be learned automatically?
7. Section 7 and Appendix C mention inverse-transform and Langevin sampling. How efficient are these in high-dimensional cases, and have you compared them to Hamiltonian Monte Carlo or other samplers for posterior recovery?
8. Since each latent basis has an interpretable contribution to the posterior, can you comment on whether the learned bases correspond to meaningful latent modes (especially in the astrophysical tasks)?

---

> ### Author Response · Authors · 2025-11-22
> **Thanks for your review**
>
> Thank you for the thoughtful review. We aim to address your comments and questions below. We will make any changes mentioned below during the discussion period.
>
> > Although Algorithm 1 provides a gradient estimator for the objective function, it lacks pseudocode for the entire algorithm. Including this would clarify the overall process.
>
> We will add pseudocode for the full algorithm.
>
> > Given that both the basis function and inference network are parameterized as deep neural networks, computational efficiency remains a concern for this algorithm…How does the computational efficiency of the proposed LBF-NPE compare to other baselines? Can you provide inference time comparisons?
>
> We have added computational cost baselines, which show that our method has a runtime comparable to competitors’. Please see the table in the revised Appendix E.7 of the paper pdf file for ease. Although $f$ and $s$ are parameterized as deep networks, in the amortized inference literature, the use of deep neural networks is standard, as it enables efficient large-scale inference. Although amortization requires training, which can be computationally expensive, inference time can be much more efficient because only a single forward pass through the fitted network is needed per data point.
>
> > While the impact of basis dimensionality and normalization is explored (App. E.1–E.3), ablations on the effect of stereographic projection and alternating optimization are somewhat fragmented…what guarantees or empirical evidence exist when both networks are trained jointly?
>
> Our main plots demonstrating the effects of stereographic parameterization are in Appendix E.1 for the spiral example. We will add more discussion around this section and reference it in the main text. Regarding alternating optimization, although marginal convexity (cf. Section 4.2) motivates this approach, we find in practice that, because parameter updates are quite small per step, jointly optimizing both $f_\phi$ and $s_\psi$ yields results that differ little from alternating optimization. Therefore, the alternating optimization routine, though viable and natural in some ways, isn’t an essential aspect of our method. We will update the main text to clarify this.
>
> > The choice of the scaling factor is ad-hoc. Is it tuned per task, or could it be learned automatically?
>
> We’ve added more discussion of the scaling factor in Appendix D.7. We view the scaling factor as a means to alleviate numerical stability concerns, by preventing vectors from growing in magnitude without bound during training. In both our experiments  (the toy examples and the astronomical example, we use the same scaling factor ($w=3$). Our results do not appear to be particularly sensitive to it.
>
> > Can you provide quantitative results comparing fixed (B-spline/wavelet) versus adaptive bases in terms of convergence speed, stability, and overfitting? How do you decide whether to use B-splines, wavelets, or neural bases across different tasks? Is there an automatic basis selection mechanism?
>
> The B-splines and wavelets are just convenient examples of bases that could be used. Any family of basis functions (e.g., Legendre, Hermite, Bernstein polynomials; Fourier bases, etc.) could be used. We don’t recommend a particular basis to practitioners because specific problems may suggest one choice over another in the fixed-basis setting. Trying multiple fixed bases and selecting the best one based on held-out log-likelihood is a viable automatic basis selection strategy. Alternatively, our adaptive neural basis functions can be used.
>
> The fixed-basis setting is more closely related to work on VI with basis expansions (e.g., EigenVI), whereas our adaptive neural basis formulation goes beyond typical VI approaches.

---

> > ### Author Response · Authors · 2025-11-22
> > **On your additional questions**
> >
> > > …stronger evidence on real higher-dimensional tasks would strengthen the claim of scalability…Could the authors elaborate on whether LBF-NPE can be extended to high-dimensional latent spaces (e.g., posteriors of Bayesian neural networks) without prohibitive integration cost? Given time constraints, you can illustrate this only using a small toy regression experiment…[and also your comment on]...Section 7 and Appendix C mention inverse-transform and Langevin sampling. How efficient are these in high-dimensional cases…
> >
> > We’re working on a toy BNN example now and aim to share the results soon. More generally on higher-dimensional settings you're asking about, the main purpose of our method is to marginalize over nuisance variables in high-dimensional problems. Ultimately, the target of our inference is typically low-dimensional even in high-dimensional settings with many latent variables.
> >
> > In the BNN setting, for example, integrating out the neural network weights $w$ is the main complication for inference. The strength of an NPE formulation is that we can directly target the posterior marginal $p(y' \mid x’)$ *during training*, rather than at inference time. Further, the marginalization is implicit: no integration needs to occur over the space $w$, only over the low-dimensional domain of scalar $y'$ to get a proper density. Our astronomical-redshift and object-detection experiments also demonstrate this phenomenon, targeting two-dimensional (location) or one-dimensional (redshift) marginals of the posterior, marginalizing over nuisance latent variables of the generative model.
> >
> > The sampling approaches in Section 7 and Appendix C are intended simply as examples of standard methods for sampling from low-dimensional density functions. Any approach, including HMC, could be used. The main value of our method lies in marginalizing over nuisance latent random variables, leaving only a low-dimensional inference problem that can be readily solved using standard methods.
> >
> > > The current gradient estimator is biased but consistent. Have you quantified this bias, or explored control variates or reparameterization tricks to reduce it?
> >
> > Because we infer low-dimensional projections of the posterior distribution, rather than the high-dimensional posterior, we haven’t found the bias to be substantial enough to affect our results. In low dimensions, many strategies can be used to reduce the bias to an arbitrarily low level. The use of control variates seems promising if we were to extend our method to high-dimensional posterior projections. However, we see the main value of our method to be in marginalizing over high-dimensional nuisance random variables, leaving only a low-dimensional inference problem.
> >
> > > Since each latent basis has an interpretable contribution to the posterior, can you comment on whether the learned bases correspond to meaningful latent modes (especially in the astrophysical tasks)?
> >
> > We’ve been thinking about this too. Indeed, we find that the learned bases can serve as templates for the posterior distributions. The fitted basis functions don’t seem to be modal themselves, at least in this example: in other words, a single basis function does not look like a typical posterior distribution. We see this most strikingly in Figures 17-20 of Appendix E.3. Rather, the basis functions appear to fit templates that can be combined to produce posteriors. In the example referenced above, these posteriors are sparse except for two local objects, unlike the bases, which are quite varied.

---

> > > ### Author Response · Authors · 2025-11-25
> > > **BNN Toy Example**
> > >
> > > Thanks again to the reviewer for this suggestion for illustration. We've added an example of a toy BNN regression problem (please scroll to the very bottom of the revised pdf, Appendix E.9). With the additional space of an extra page, we would aim to discuss this example in the main text.
> > >
> > > Given a small conditioning dataset $D$ of noisy observations, LBF-NPE directly targets the posterior predictive $p(y' \mid x', D)$, amortizing over arbitrary $x'$. This is a low-dimensional marginal of the full posterior, marginalizing over the weights of the neural network, a high dimensional set of nuisance latent variables in this problem. Despite the latent space being high-dimensional, LBF-NPE can fit the marginal without computing an explicit integral over the weights. LBF-NPE accomplishes this by directly targeting the marginal $p(y' \mid x', D)$ by construction during training, rather than at inference time. In this way, we can fit the low-dimensional approximate posterior of interest without prohibitive cost of integration over neural network weights.

---

### Official Review · Reviewer_L4UB · 2025-11-01

**Soundness:** 3
**Presentation:** 3
**Contribution:** 3
**Rating:** 6
**Confidence:** 2

**Summary:**

This paper introduces LBF-NPE, a neural posterior estimation method that parameterizes variational distributions using latent basis expansions. The method models log-densities as linear combinations of basis functions, forming an exponential family that balances expressivity with optimization stability. The authors propose several variants, including fixed basis functions and adaptive basis learning with stereographic projection for identifiability.

**Strengths:**

* Novel variational family that bridges the gap between simple distributions and complex black-box models.
* Strong theoretical foundation with convexity guarantees in certain settings.
* Comprehensive experiments across synthetic and real-world scientific problems.
* Outperforms MDNs and other baselines on multiple benchmarks.
* Effective handling of multimodal and complex posterior geometries, as demonstrated by the experiments.

**Weaknesses:**

* There seems to be no discussion of computational cost compared to baselines.
* If experiments are repeated several times with different random seeds, error bars should be reported in the plot and tables.

**Questions:**

* How to determine the number of samples and the number of basis $K$, and how sensitive is the performance to them?

---

> ### Author Response · Authors · 2025-11-22
> **Thank you for your review**
>
> Thank you very much for the review. We would like to address the main questions you bring up.
>
> > There seems to be no discussion of computational cost compared to baselines.
>
> We have performed additional runs to evaluate the computational costs of the methods we evaluate against. Please see the table in Appendix E.7 of the paper pdf, which we have updated. All methods are roughly comparable in computational cost. Although LBF-NPE is slightly more expensive per step, it converges in fewer steps and, at worst, incurs a 2x slowdown in the experiments we consider.
>
> > How to determine the number of samples and the number of basis, and how sensitive is the performance to them?
>
> Generally, we find that using more basis functions helps because they increase expressivity. We have some ablations showing this in Table 4 (Appendix E.3), which demonstrates that the KL (both forward and reverse) decreases as more basis functions are added. We will move this ablation and discussion of this topic to the main text with the additional page of space permitted.
>
> We generally recommend using as many basis functions as available computational resources will allow. We expect a threshold beyond which the performance gain from including additional basis functions is minimal and does not warrant additional computation. The user can vary the number of basis functions and examine metrics on a holdout set for each, looking for the number beyond which any gain is marginal.
>
> > If experiments are repeated several times with different random seeds, error bars should be reported in the plot and tables.
>
> Thanks for this suggestion. We have added error bars where relevant to our experiments. Please find the revised tables given Appendix E.8 (please scroll to the bottom of the pdf). We will move these to the main text in our final revision.

---

### Official Review · Reviewer_ZrDB · 2025-11-01

**Soundness:** 4
**Presentation:** 3
**Contribution:** 3
**Rating:** 8
**Confidence:** 3

**Summary:**

This paper proposes a variational family LBF-NPE that is specialized for neural posterior estimation, which is a simulation-based (i.e., likelihood-free) inference method of the Bayesian posterior. Specifically, the variational family is an exponential family where the natural parameter $\eta$ is given by a function $f_{\phi}(x)$, providing amortization across observations $x$, and the sufficient statistic is a vector of $K$ basis functions $s_{\psi}(z)$. This means the optimization depends only on the inner product between $f_{\phi}(x)$ and $s_{\psi}(z)$, leading to several properties, including affine gradients, convexity in $f$ and $s$, and the ability to fix the basis functions. Redundancy between $\phi$ and $\psi$ can be mitigated by a stereographic projection approach. The paper provides four experiments on four datasets, two are low dimensional toys and two are more realistic datasets related to astronomy, showing improvements in NLL and/or KL. The paper includes detailed explanations of the experiments in the appendix.

**Strengths:**

Overall this is a well-written and thorough paper that provides theoretical and empirical backing for the proposed method. Given recent interest in basis-expansion methods in VI (e.g., eigenVI), I think it provides a nice contribution. It’s interesting that LBF-NPE performs better than eigenVI with fewer basis functions. I appreciate the attempt at a middle-ground between, say, mixtures of Gaussians and normalizing flows. This gives the user a few levers (e.g., fixing the basis) to ease the complexity of the optimization problem. The experiments showed improvements over recent baselines.

**Weaknesses:**

- The paper aims to strike a balance between simplicity and expressivity in the variational family. However, since the amortization $f_\phi$ is a neural network in parameters $\phi$, one concern is that it doesn’t make the optimization problem that much easier (see my question below regarding the importance of convexity in the function $f$).
- As the authors point out, sampling from the variational distribution in LBF-NPE is challenging.
- The results of the object detection experiment are buried in the appendix. As I understand it, the takeaway is that more basis functions reduces the KL to the chosen target but at a diminishing rate.

**Questions:**

- Regarding sections 4.1 and 4.2, can you comment on the importance of convexity in $f$ and $s$ given that they are potentially nonlinear functions of $\phi$ and $\psi$, respectively? For example, if you fix the basis functions, then the objective in Eq 6 may still be nonconvex in $\phi$ (e.g., if $f_\phi$ is a neural network), is that correct? If so, can you expand on how being convex in $f_\phi$  but not $\phi$ is still helpful?
- In the Discussion & Limitations section, you write that regions of the latent space can be “zeroed-out”. I understand this to mean that some of the coefficients $\eta = f_\phi$ are zero after training. Can you comment on why this is useful and which experiments show this behavior? It’s interesting this can happen without anything encouraging sparsity.

---

> ### Author Response · Authors · 2025-11-22
> **Thank you for your review**
>
> Thank you very much for the detailed and encouraging review.
>
> > since the amortization $f_\phi$ is a neural network in parameters $\phi$, one concern is that it doesn’t make the optimization problem that much easier...For example, if you fix the basis functions, then the objective in Eq 6 may still be nonconvex in $\phi$ (e.g., if $f_\phi$ is a neural network), is that correct? If so, can you expand on how being convex in $f_\phi$ but not $\phi$ is still helpful?
>
> One benefit of our formulation is that it directly inherits the convexity results of previous work on overparameterized networks ([1],[2]). This line of analysis, begun with the work on the neural tangent kernel ([1]), directly relates optimization of neural networks in parameter space to the properties of the loss function $L(f)$ being optimized in functional space. The key idea is that as neural networks grow arbitrarily wide (and thus arbitrarily flexible), the solutions $f$ found by gradient descent mirror those obtained by optimizing $L(f)$ directly – hence, the benefit of convexity of $L(f)$ for convergence to a global minimizer.
>
> This result is surprising, and perhaps unintuitive, as neural network optimization landscapes in parameter space remain nonconvex, as you point out. Results in this line of work tend to only formally hold in the asymptotic limit of infinitely wide networks, although practical analysis remains an ongoing line of research. We elaborate more on this line of work and its relation to our results in Appendix B. The key takeaway for our work, which applies these results, is that finite-width, applied settings, while they may lack the same formal guarantees as the infinitely-wide limit, often inherit beneficial behavior due to the convexity of the functional formulation. We show (Section 6.1) that our convex formulation helps avoid local minima issues compared to MDNs. Although both approaches will have local minima in $\phi$, our plots examine the quality of the solution $f$. Our formulation converges (approximately) to a global optimum in $f$ reliably (which one typically cares about when evaluating the method). The experiments in [2] provide further evidence for this in the finite-width setting.
>
>
> [1] Neural Tangent Kernel: Convergence & Generalization in Neural Networks (Jacot et al.)
>
> [2] Globally Convergent Variational Inference (McNamara et al.)
>
> > The results of the object detection experiment are buried in the appendix. As I understand it, the takeaway is that more basis functions reduces the KL to the chosen target but at a diminishing rate.
>
> That’s correct. For this experiment, we show that KL decreases with the number of basis functions. More broadly, though, we wanted to demonstrate, practically, the success of this experiment by varying the number of basis functions. For localizing objects that can be placed anywhere in the image, it’s not immediately clear how the fitted basis functions, which are globally defined, achieve this; the form of the basis functions for $k=9,..20$ makes this explicit. We will add more discussion of the implications of these experiments to the text.
>
> > In the Discussion & Limitations section, you write that regions of the latent space can be “zeroed-out”. I understand this to mean that some of the coefficients are zero after training. Can you comment on why this is useful and which experiments show this behavior? It’s interesting this can happen without anything encouraging sparsity.
>
> Our phrasing here was too colloquial; we’ll revise it. The intended point was that the linear combination of basis functions $\eta^\top b(\theta)$ can take on negative values in our construction, since coefficients or adaptive basis functions can be negative. This contrasts with other basis-function VI methods (e.g., EigenVI), which have strictly positive coefficients and positive basis functions.
>
> When forming the density of our variational distribution, these negative values are exponentiated and normalized (similar to a softmax), resulting in extremely small values. Thus, the density in these regions is nearly zero, even though it is not identically zero.
>
> Our variational family has the flexibility to accurately approximate posteriors with sparse support (as is the case for object detection with objects that can be localized). If the target density is highly peaked (most mass in a small region), our parameterization can achieve this provided $\eta^\top b(\theta) \approx -20$, for example, in the region with negligible mass. After exponentiating, $\exp (-20) \approx 0$.

---

### Meta-Review · Area_Chair_SeTd · 2025-12-12

**Summary:**

Overall, the reviewers found the paper interesting and valuable, as it directly models the log density of the variational distributions as a linear combination of basis functions, which offers benefits such as unconstrained optimization (easier to optimize than constrained ones) and a balance between expressiveness and simplicity.

Most of the reviewers' concerns are appropriately addressed. The authors also addressed a remaining issue (described below) raised by reviewer hPZ3.

Whether the final reviewer increased the score or not, I think this paper is in good shape and clearly delivers core contributions. Hence, I recommend accepting this paper.

**Reviewer Concerns:**

Most of the reviewers' concerns are appropriately addressed.

The remaining one is *"Most experiments are in ≤ 2D latent spaces. The high-dimensional example (App. E.4) is also synthetic (50-dimensional); stronger evidence on real higher-dimensional tasks would strengthen the claim of scalability." by **reviewer hPZ3***.
The authors performed a new toy experiment to rebut, but I am not 100% sure if this would have satisfied the reviewer, as I do not know what high-dimensional this reviewer could mean (high or low dimension could be relative). From my point of view, this new toy experiment still works well with **higher**-dimensional latent spaces than the other experiments in the originally submitted paper, and so this is satisfying.

An additional note: I would also be interested in knowing how this method would perform under the models with high-dimensional targets.

**Reviewer Scores:**

The first two reviewers' scores would likely remain the same. The question is whether the last reviewer (hPZ3) would have changed the score based on how satisfactory the new toy experiment and the authors' reply are to the reviewer.

---

### Decision · Program_Chairs · 2026-01-26

Accept (Poster)